# Sparse Contextual CDF Regression

**Kamyar Azizzadenesheli**                                          *kamyara@nvidia.com*
*Nvidia Corporation*

**William Lu**                                                      *lu909@purdue.edu*
*Department of Computer Science*
*Purdue University*

**Anuran Makur**                                                    *amakur@purdue.edu*
*Department of Computer Science and School of Electrical and Computer Engineering*
*Purdue University*

**Qian Zhang**                                                      *zhan3761@purdue.edu*
*Department of Statistics*
*Purdue University*

**Reviewed on OpenReview:** *https://openreview.net/forum?id=AIc48TjuSt*

## Abstract

Estimating cumulative distribution functions (CDFs) of context-dependent random variables is a central statistical task underpinning numerous applications in machine learning and economics. In this work, we extend a recent line of theoretical inquiry into this domain by analyzing the problem of *sparse contextual CDF regression*, wherein data points are sampled from a convex combination of $s$ context-dependent CDFs chosen from a set of $d$ basis functions. We show that adaptations of several canonical regression methods serve as tractable estimators in this functional sparse regression setting under standard assumptions on the conditioning of the basis functions. In particular, given $n$ data samples, we prove estimation error upper bounds of $\tilde{O}(\sqrt{s/n})$ for functional versions of the lasso and Dantzig selector estimators, and $\tilde{O}(\sqrt{s}/\sqrt[4]{n})$ for a functional version of the elastic net estimator. Our results match the corresponding error bounds for finite-dimensional regression and improve upon CDF ridge regression which has $\tilde{O}(\sqrt{d/n})$ estimation error. Finally, we obtain a matching information-theoretic lower bound which establishes the minimax optimality of the lasso and Dantzig selector estimators up to logarithmic factors.

## 1 Introduction

The estimation of cumulative distribution functions (CDFs) is a classical problem in mathematical statistics stemming back to the Glivenko-Cantelli theorem (Cantelli, 1933; Glivenko, 1933; Devroye et al., 2013), which states that empirical CDFs constructed from independent samples of a single random variable converge uniformly to the random variable's true CDF. Subsequent classical research in this area has focused on deriving tight non-asymptotic sample complexity results in terms of the Kolmogorov-Smirnov distance among others, such as the Dvoretzky-Kiefer-Wolfowitz inequality (Dvoretzky et al., 1956) and improved bounds by Massart (1990).

Motivated by applications to modern learning tasks such as contextual bandits and Markov decision processes (Huang et al., 2021; 2022), a recent line of research (Zhang et al., 2024) introduced the problem of *context-dependent* CDF estimation, which requires a learner to simultaneously estimate a (possibly infinite) family of CDFs parameterized by some context variable. As an initial simplification, the authors considered

---

The author ordering is alphabetical.

the restricted setting of *contextual CDF regression*, wherein the true contextual CDF is a convex combination of $d$ context-dependent basis functions. The authors generalized the classical ridge regression method (Hoerl & Kennard, 1970; Abbasi-Yadkori et al., 2011) to this functional regression problem and derived a tight $\tilde{O}(\sqrt{d/n})$ estimation error bound given $n$ samples in a variety of data generation settings. However, when $d$ is large, specifically in unstructured settings where a massive set of potential CDFs are considered, ridge regression utilizes all CDF basis functions for the purpose of estimation without regard to their relevance.

In this paper, as a further step towards developing general algorithms for contextual CDF estimation, we propose *sparse regression* and *basis selection* techniques for the aforementioned CDF regression problem based on functional versions of lasso (Tibshirani, 1996), elastic net (Zou & Hastie, 2005), and Dantzig selector (Candes & Tao, 2007) methods. Crucially, all of our techniques achieve estimation bounds with no polynomial dependence on $d$, allowing accurate recovery of the true contextual CDF from bases containing exponentially many irrelevant functions. We also establish minimax optimality results for sparse CDF regression.

## 1.1 Outline

We briefly delineate the structure of our paper. We introduce the sparse contextual CDF regression problem under present investigation in Section 1.2 and define relevant notation in Appendix A. In Section 2, we outline some applications of CDFs in machine learning and economics, and provide an overview of related functional regression schemes in the previous scientific literature. In Section 3, we state our main contributions and discuss the overarching proof techniques used throughout the paper. Section 4 provides the formal derivation of our upper bound on lasso estimation error in the fixed design setting. We include numerical simulations of our data generation and parameter estimation processes in Section 5. We defer remaining proofs and technical details to Appendices B to E, provide examples of CDF bases which satisfy the preconditions for our main results in Appendix F, and briefly discuss the computational complexity of our estimators in Appendix G.

## 1.2 Model and Setup

In this subsection, we formally define the sparse contextual CDF regression problem under investigation in this paper. (Recall that we utilize the notation defined in Appendix A.) Let $\mathcal{X}$ denote a general context space. We refer to a function $f(x, t) : \mathcal{X} \times \mathbb{R} \to [0, 1]$ as a *contextual CDF* if for any $x \in \mathcal{X}$, $f(x, \cdot)$ is a valid CDF for a real-valued random variable with range contained in some set $S \subseteq \mathbb{R}$. Let $\mathfrak{m}$ be any probability measure on $S$. Let $\{\phi_1, \ldots, \phi_d\}$ denote a fixed basis of $d \in \mathbb{N}$ contextual CDFs indexed by $i \in [d]$. For convenience, we often conceptualize this basis as a single vector-valued function $\Phi : \mathcal{X} \times \mathbb{R} \to [0, 1]^d$ defined by $[\Phi(x, t)]_i = \phi_i(x, t)$ for all $i \in [d]$.

Let $F : \mathcal{X} \times \mathbb{R} \to [0, 1]$ denote the true contextual CDF we aim to recover. Following the precedent established in Zhang et al. (2024), we assume that $F$ is a convex combination of the basis functions $\{\phi_1, \ldots, \phi_d\}$. Hence, the problem reduces to recovering the true parameter vector $\theta_* \in \Delta^{d-1}$ such that

$$\forall x \in \mathcal{X}, \, \forall t \in \mathbb{R}, \,\, F(x, t) = \theta_*^\top \Phi(x, t) \,.$$

Let $S_* = \mathrm{supp}(\theta_*)$ denote the support of the true parameter. For notational simplicity, let $s = |S_*| = \|\theta_*\|_0$. Let $\{(x^{(1)}, y^{(1)}), \ldots, (x^{(n)}, y^{(n)})\}$ be a set of $n \in \mathbb{N}$ observed samples indexed by $j \in [n]$, generated according to the *fixed design* or *random design* settings described below:

- **Fixed design**. For each $j \in [n]$, the context variable $x^{(j)} \in \mathcal{X}$ is fixed a priori, and the response variable $y^{(j)} \in \mathbb{R}$ is independently sampled from the CDF $F(x^{(j)}, \cdot)$.

- **Random design**. For each $j \in [n]$, the context variable $x^{(j)} \in \mathcal{X}$ is independently sampled from an unknown probability distribution $P_X^{(j)}$ over $\mathcal{X}$. Then, conditional on $x^{(j)}$, the response variable $y^{(j)} \in \mathbb{R}$ is independently sampled from the CDF $F(x^{(j)}, \cdot)$.

These design settings extend the data generation processes described in Abbasi-Yadkori et al. (2011) and Hsu et al. (2012). Let $\Phi_j(t) = \Phi(x^{(j)}, t)$ denote the CDF basis at the $j$th context variable, and let $x^{1:n} =$

$(x^{(1)}, \ldots, x^{(n)})$ denote a collection of sampled variables, with analogous definitions for $y^{1:n}$ and $(x, y)^{1:n}$. Let $U_n \in \mathbb{R}^{d \times d}$ denote the *empirical n-sample Gramian matrix* given by

$$\forall i, i' \in [d], \ [U_n]_{i,i'} = \frac{1}{n} \sum_{j=1}^{n} \left\langle [\Phi_j]_i, [\Phi_j]_{i'} \right\rangle \qquad \qquad \therefore U_n = \frac{1}{n} \sum_{j=1}^{n} \int_S \Phi_j \Phi_j^\top \, d\mathfrak{m}. \tag{1}$$

In the random design setting, let $\Sigma_n \in \mathbb{R}^{d \times d}$ denote the *expected n-sample Gramian matrix* given by

$$\forall i, i' \in [d], \ [\Sigma_n]_{i,i'} = \frac{1}{n} \sum_{j=1}^{n} \mathop{\mathbb{E}}_{X^{(j)} \sim P_X^{(j)}} \left[ \left\langle [\Phi_j]_i, [\Phi_j]_{i'} \right\rangle \right] \qquad \qquad \therefore \Sigma_n = \mathop{\mathbb{E}}_{X^{1:n}} \left[ \frac{1}{n} \sum_{j=1}^{n} \int_S \Phi_j \Phi_j^\top \, d\mathfrak{m} \right]. \tag{2}$$

As mentioned previously, the objective of contextual CDF regression is to recover $\theta_*$ given the samples $(x, y)^{1:n}$. Each $Y^{(j)}$ defines a one-sample empirical CDF $\mathrm{I}_{Y^{(j)}}(t) = \mathbb{1}\{t \geq Y^{(j)}\}$ which approximates the true CDF $F(x^{(j)}, \cdot) = \mathbb{E}[\mathrm{I}_{Y^{(j)}}(\cdot) \mid X^{(j)} = x^{(j)}]$ in expectation conditioned on $X^{(j)} = x^{(j)}$. We investigate three estimators for $\theta_*$ based on the paradigm of empirical risk minimization. Firstly, the *lasso estimator* $\hat{\theta}_\lambda$ imposes an $\ell^1$-penalty on the parameter vector, weighted by $\lambda > 0$:

$$\hat{\theta}_\lambda = \arg \min_{\theta \in \mathbb{R}^d} \left\{ \frac{1}{n} \sum_{j=1}^{n} \left\| \mathrm{I}_{y^{(j)}} - \theta^\top \Phi_j \right\|_{\mathcal{L}^2(S, \mathfrak{m})}^2 + \lambda \|\theta\|_1 \right\}. \tag{3}$$

Secondly, the *elastic net estimator* $\hat{\theta}_{\lambda_1, \lambda_2}$ imposes both $\ell^1$- and $\ell^2$-penalties on the parameter vector, weighted by $\lambda_1 > 0$ and $\lambda_2 > 0$ respectively:

$$\hat{\theta}_{\lambda_1, \lambda_2} = \arg \min_{\theta \in \mathbb{R}^d} \left\{ \frac{1}{n} \sum_{j=1}^{n} \left\| \mathrm{I}_{y^{(j)}} - \theta^\top \Phi_j \right\|_{\mathcal{L}^2(S, \mathfrak{m})}^2 + \lambda_1 \|\theta\|_1 + \lambda_2 \|\theta\|^2 \right\}. \tag{4}$$

Thirdly, the *Dantzig selector* $\bar{\theta}_\lambda$ is given by the following variation on the constrained optimization formulation of lasso estimation:

$$\bar{\theta}_\lambda = \underset{\theta \in \mathbb{R}^d}{\text{minimize}} \ \|\theta\|_1 \tag{5}$$

$$\text{subject to} \ \left\| \frac{1}{n} \sum_{j=1}^{n} \left\langle \mathrm{I}_{y^{(j)}} - \theta^\top \Phi_j, \Phi_j \right\rangle \right\|_\infty \leq \lambda. \tag{6}$$

For all three estimators, we analyze the *estimation error* given by the Euclidean distance $\|\hat{\theta} - \theta_*\|_A$ between the estimated and true parameter vectors, possibly weighted by some matrix $A$. We remark that Equations (3) to (5) define *improper estimators* which do not necessarily lie in $\Delta^{d-1}$ and are thus not guaranteed to define valid CDFs. However, since $\theta_* \in \Delta^{d-1}$ and $\Delta^{d-1}$ is closed and convex, any upper bound on the estimation error of an improper estimator also holds for its projection onto $\Delta^{d-1}$, by Beck (2014, Equation 9.10).

Lastly, we remark that our results hold for any $s \leq d$, although we are principally interested in the advantages of our proposed methods over the ridge estimator baseline (Zhang et al., 2024, Section 3.1) for small values of $s$. We emphasize that our results have no dependence on the cardinality of the context space $\mathcal{X}$.

## 2 Related Work

Broadly speaking, CDFs underpin the computation of risk functionals which inform decision-making in mathematical finance and actuarial science. For example, generic law invariant risk functions such as conditional value-at-risk (Rockafellar et al., 2000; Artzner et al., 1999) are parameterized by CDFs, and a notion of distortion risk measure (Wirch & Hardy, 2001) arises from the composition of a CDF with a distortion function. Coherent risk measures are instrumental in portfolio management and optimization (Krokhmal, 2007) and can be formulated in terms of CDFs (Shapiro et al., 2014). Lastly, spectral risk

measures are computed as weighted averages of outcomes (Acerbi, 2002) and hence depend on the entire trajectory of the underlying random variable's CDF.

Similar considerations exist in further scopes of learning theory, where the aforementioned risk functionals are incorporated into supervised learning tasks (Liu et al., 2022) and multi-armed bandit problems (Cassel et al., 2023) to model fairness, risk aversion, and distribution shift (Wong et al., 2022). The recently proposed off-policy risk assessment framework (Huang et al., 2021) includes CDF estimation as a key building block and has been applied to contextual bandit problems and Markov decision processes (Huang et al., 2022). Other work in this vein has employed the mean-variance (Sani et al., 2013; Zimin et al., 2014) and value-at-risk (Vakili & Zhao, 2015) paradigms in the multi-armed bandit setting to analyze risk-reward trade-offs. Furthermore, cumulative prospect theory is intimately tied to risk distortion and the ensuing dependence on CDF estimation (Prashanth et al., 2016), and has found relevance in reinforcement learning and stochastic optimization (Jie et al., 2018).

We touch on the well-established precedent in the scientific community for framing function estimation through the lens of linear regression and basis selection. Romo et al. (2013) generalized lasso variable selection to linear models with scalar regressors and functional responses to develop interpretable analyses of car accident data. A related method exploiting feature sparsity and output function smoothness was proposed in Barber et al. (2017) to perform genome-wide association studies. Linear models with functional regressors and scalar responses have been used in genomics, MRI data analysis, and chemometrics, spurring the development of functional group-lasso (Pannu & Billor, 2017) and wavelet-based lasso methods (Zhao et al., 2012; 2015) for such models. Prior studies on genetic regulatory networks (Hong & Lian, 2011) have incorporated function-on-function regression models with $\ell^1$-regularization to encode sparsity in pairwise interactions between genes. Lastly, recent developments on lasso estimation for function-on-function regression (Centofanti et al., 2022; Maranzano et al., 2023) find downstream application in geostatistical models and mortality data analysis.

Compared to the prior literature, the main novelty of our contributions lies in the simultaneous estimation of an entire family of CDFs. In this sense, our work may be interpreted as a generalization of the canonical mixture model with known basis distributions (Murphy, 2012, Section 11.2) to the context-dependent setting. For maximum generality, we focus on CDF estimation instead of PDFs or quantiles which require restrictions for existence and well-definedness. We formulate estimation through the lens of functional linear regression and show that our task reduces to integral computation and finite-dimensional optimization, making our approach computationally efficient using existing numerical methods. Lastly, our derivations utilize intrinsic properties of CDFs instead of performing discretization to avoid introducing approximation error, in contrast to the approach taken in Hong & Lian (2011) among others.

## 3 Main Results

### 3.1 Lasso

Before presenting our main contributions, we introduce the following condition on symmetric matrices which is used in the statements of our main results:

**Definition 1** (Restricted eigenvalue condition)**.** *Fix $d \in \mathbb{N}$, $\kappa \geq 0$, $\gamma \geq 0$, and $S_* \subseteq [d]$. Let*

$$C_\gamma(S_*) = \left\{ v \in \mathbb{R}^d : \left\| [v]_{S_*^c} \right\|_1 \leq \gamma \left\| [v]_{S_*} \right\|_1 \right\} .$$

*A symmetric matrix $A \in \mathbb{R}^{d \times d}$ satisfies the $(\kappa, \gamma)$-restricted eigenvalue condition over $S_*$ iff*

$$\forall v \in C_\gamma(S_*), \ v^\top A v \geq \kappa \left\| v \right\|^2 .$$

Intuitively, the restricted eigenvalue condition specializes the notions of positive definiteness and strong convexity to the subset of directions whose support is close to $S_*$. The significance of this condition lies in its application to the Gramian matrices $U_n$ and $\Sigma_n$, since lower-bounding the Gramian eigenvalues is sufficient to ensure the well-conditioning of the basis functions comprising the true CDF. Hence, our work instantiates

the standard definition from Wainwright (2019, Definition 7.12) on the general inner product spaces used in the definitions of $U_n$ and $\Sigma_n$. We provide examples of non-trivial CDF bases which satisfy the restricted eigenvalue condition in Appendix F.

Under this assumption, we state our first main result, a high-probability upper bound on the error of the lasso estimator in the fixed design setting:

**Theorem 1** (Lasso fixed design upper bound). *Fix $\delta \in (0,1)$ and $\kappa > 0$. Assume the samples $(x,y)^{1:n}$ are generated according to the fixed design setting. Assume $U_n$ satisfies the $(\kappa, 3)$-restricted eigenvalue condition over $S_*$. Let $\hat{\theta}_\lambda$ be the lasso estimator (3) with regularization hyperparameter $\lambda = 4\sqrt{(2/n)\log(2d/\delta)}$. Then with probability at least $1 - \delta$, the estimation error satisfies the bounds*

$$\left\|\hat{\theta}_\lambda - \theta_*\right\|_{U_n} \le 6\sqrt{\frac{2s}{\kappa n}\log\left(\frac{2d}{\delta}\right)} \quad and \quad \left\|\hat{\theta}_\lambda - \theta_*\right\| \le \frac{6}{\kappa}\sqrt{\frac{2s}{n}\log\left(\frac{2d}{\delta}\right)}.$$

We provide the technical details of our proof in Section 4. The crux of our argument lies in the definition of $\hat{\theta}_\lambda$ as the minimizer of the empirical risk objective in (3). Substituting $\hat{\theta}_\lambda$ and $\theta_*$ into this objective results in an inequality (15) in terms of $\hat{\theta}_\lambda$ and $\theta_*$. Rearranging yields an upper-bound (17) on the estimation error $\|\Delta\|_{U_n}$ in terms of two quantities:

- The scalar projections of the sampling errors $\mathrm{I}_{y^{(j)}} - \theta_*^\top \Phi_j$ onto the $d$ basis functions $\Phi_j$, which we upper-bound with high probability (10) using Hoeffding's inequality and the union bound over $i \in [d]$.

- The difference between the $\ell^1$-regularizers $\|\theta_*\|_1 - \|\hat{\theta}_\lambda\|_1$, which we upper-bound by utilizing the sparsity of $\theta_*$ and the definition of $\ell^1$-norm.

Combining these results gives rise to a bound on the relative magnitudes of the components of $\Delta$ corresponding to the true support $S_*$ and its complement (21). Then, upper-bounding the component $\|[\Delta]_{S_*}\|_1$ is sufficient (22) to characterize the overall estimation error $\|\Delta\|_{U_n}$. Applying various norm equivalences and invoking the restricted eigenvalue condition on $U_n$ completes the proof. We remark that the restricted eigenvalue condition plays a comparable role in ensuring the well-conditioning of the Gramian matrix $U_n$ in our work, as it does for $X^T X$ in classical formulations of finite-dimensional regression methods.

Theorem 1 establishes the asymptotic complexity of lasso CDF regression as $O(\sqrt{s\log(d)/n}) = \tilde{O}(\sqrt{s/n})$, in direct analogy to the $\tilde{O}(\sqrt{d/n})$ result for ridge regression in Zhang et al. (2024). The dependence on $\sqrt{s}$ arises from our sparsity analysis on $\|\theta_*\|_1 - \|\hat{\theta}_\lambda\|_1$, which reduces the problem to bounding $\|[\Delta]_{S_*}\|_1$ as discussed above. The factor of $1/\sqrt{n}$ is a consequence of Hoeffding's inequality, and the factor of $\sqrt{\log(d)}$ arises from the union bound over $i \in [d]$ in our analysis of the sampling errors. Ultimately, the lasso estimator's characteristic $\ell^1$-penalty term plays an instrumental role in deriving a result with sub-polynomial dependence on $d$.

Our second main result is a high-probability upper bound on the error of the lasso estimator in the random design setting, under similar assumptions on the well-conditioning of the CDF basis:

**Theorem 2** (Lasso random design upper bound). *Fix $\delta_1 \in (0,1)$, $\delta_2 \in (0, 1 - \delta_1)$, and $\kappa > 0$. Assume the samples $(x,y)^{1:n}$ are generated according to the random design setting. Assume that for any $n \in \mathbb{N}$, the matrix $\Sigma_n$ satisfies the $(\kappa, 3)$-restricted eigenvalue condition over $S_*$. Assume the sample size is at least $n \ge (32d^2/\kappa^2)\log(d/\delta_1)$. Let $\hat{\theta}_\lambda$ be the lasso estimator (3) with regularization hyperparameter $\lambda = 4\sqrt{(2/n)\log(2d/\delta_2)}$. Then with probability at least $1 - \delta_1 - \delta_2$, the estimation error satisfies the bounds*

$$\left\|\hat{\theta}_\lambda - \theta_*\right\|_{U_n} \le 12\sqrt{\frac{s}{\kappa n}\log\left(\frac{2d}{\delta_2}\right)} \quad and \quad \left\|\hat{\theta}_\lambda - \theta_*\right\| \le \frac{12}{\kappa}\sqrt{\frac{2s}{n}\log\left(\frac{2d}{\delta_2}\right)}.$$

We defer the technical details of our proof to Appendix B. In a nutshell, since $U_n$ is a sum of $n$ independent matrices and is an unbiased estimator of $\Sigma_n$, we can employ a matrix analogue of Hoeffding's inequality (23) to justify approximating $\Sigma_n$ with $U_n$ given sufficiently many samples $n$. This reduces the problem to the fixed design setting, and invoking Theorem 1 completes the proof.

## 3.2 Elastic Net

Our third main result is a high-probability upper bound on the error of the elastic net estimator in the fixed design setting:

**Theorem 3** (Elastic net fixed design upper bound). *Fix $\delta \in (0,1)$. Assume the samples $(x,y)^{1:n}$ are generated according to the fixed design setting. Assume $U_n$ satisfies the $(\kappa, 3 + 4\lambda_2/\lambda_1)$-restricted eigenvalue condition over $S_*$. Let $\hat{\theta}_{\lambda_1,\lambda_2}$ be the elastic net estimator (4) with $\ell^1$-regularization hyperparameter $\lambda_1 = 4\sqrt{(2/n)\log(2d/\delta)}$. Then with probability at least $1 - \delta$, the estimation error satisfies the bounds*

$$\left\| \hat{\theta}_{\lambda_1,\lambda_2} - \theta_* \right\|_{U_n + \lambda_2 I_d} \le \left( 6\sqrt{\frac{2}{n} \log\left(\frac{2d}{\delta}\right)} + 2\lambda_2 \right) \sqrt{\frac{s}{\kappa + \lambda_2}}, \tag{7}$$

$$\left\| \hat{\theta}_{\lambda_1,\lambda_2} - \theta_* \right\| \le \left( 6\sqrt{\frac{2}{n} \log\left(\frac{2d}{\delta}\right)} + 2\lambda_2 \right) \frac{\sqrt{s}}{\kappa + \lambda_2}.$$

*Furthermore, if $\kappa < (3/2)\sqrt{(2/n)\log(2d/\delta)}$ and $\lambda_2 = 3\sqrt{(2/n)\log(2d/\delta)} - 2\kappa$, the bound (7) implies*

$$\left\| \hat{\theta}_{\lambda_1,\lambda_2} - \theta_* \right\|_{U_n + \lambda_2 I_d} \le 4\sqrt{s} \left( 3\sqrt{\frac{2}{n} \log\left(\frac{2d}{\delta}\right)} - \kappa \right)^{\frac{1}{2}} \tag{8}$$

$$\le 4 \sqrt[4]{\frac{18s^2}{n} \log\left(\frac{2d}{\delta}\right)}.$$

We emphasize that Theorem 3 produces non-trivial estimation bounds even when no assumptions are placed on $U_n$ (i.e., $\kappa = 0$), in which case the bound in (7) has $\tilde{O}(\sqrt{s}/\sqrt[4]{n})$ complexity. However, when $\kappa > 0$ and the $\ell^2$-regularization hyperparameter is sufficiently small with $\lambda_2 = O(1/\sqrt{n})$, the elastic net estimation bound exhibits $\tilde{O}(\sqrt{s/n})$ scaling, comparable with the result for the lasso estimator in Theorem 1.

At a high level, the elastic net estimator with its characteristic $\ell^2$-penalty term may be perceived as performing lasso regression on the "regularized" Gramian matrix $U_n + \lambda_2 I_d$, whose smallest eigenvalue is strictly positive. Additional conditions on $U_n$ further tighten the estimation bounds by bolstering the minimum eigenvalue of $U_n + \lambda_2 I_d$. Hence, the elastic net estimator simultaneously selects features and regularizes the problem in an integrated fashion by ensuring strong convexity of the objective function (4).

We defer the technical details of our proof to Appendix C. Our arguments mirror the proof structure for Theorem 1, with adjustments to accommodate the elastic net estimator's $\ell^2$-penalty term. At the outset, substituting $\hat{\theta}_{\lambda_1,\lambda_2}$ and $\theta_*$ into the empirical risk objective (4) and rearranging yields an upper-bound on $\|\Delta\|_{U_n + \lambda_2 I_d}$ instead of $\|\Delta\|_{U_n}$. The resulting expression (32) contains an additional quantity $\lambda_2 \Delta^\top \theta_*$, which we upper-bound in terms of the error component $\|[\Delta]_{S_*}\|_1$ in (34). Combining this contribution with the terms inherited from the lasso objective gives rise to the parenthesized sum in (7), and choosing the optimal value of $\lambda_2$ to balance the summands achieves the bound in (8).

By the same technique based on matrix Hoeffding that we introduced in Section 3.1, we also obtain the following high-probability upper bound on the error of the elastic net estimator in the random design setting, which we prove in Appendix C:

**Theorem 4** (Elastic net random design upper bound). *Fix $\delta_1 \in (0,1)$, $\delta_2 \in (0, 1 - \delta_1)$, and $\kappa > 0$. Assume the samples $(x,y)^{1:n}$ are generated according to the random design setting. Assume that for any $n \in \mathbb{N}$, the matrix $\Sigma_n$ satisfies the $(\kappa, 3 + 4\lambda_2/\lambda_1)$-restricted eigenvalue condition over $S_*$. Assume the sample size is at least $n \ge (32d^2/\kappa^2)\log(d/\delta_1)$. Let $\hat{\theta}_{\lambda_1,\lambda_2}$ be the elastic net estimator (4) with $\ell^1$-regularization hyperparameter $\lambda_1 = 4\sqrt{(2/n)\log(2d/\delta_2)}$. Then with probability at least $1 - \delta_1 - \delta_2$, the estimation error satisfies the bounds*

$$\left\| \hat{\theta}_{\lambda_1,\lambda_2} - \theta_* \right\|_{U_n + \lambda_2 I_d} \le \left( 6\sqrt{\frac{2}{n} \log\left(\frac{2d}{\delta_2}\right)} + 2\lambda_2 \right) \sqrt{\frac{2s}{\kappa + 2\lambda_2}}, \tag{9}$$

$$\left\|\hat{\theta}_{\lambda_1,\lambda_2} - \theta_*\right\| \leq \left(6\sqrt{\frac{2}{n}\log\left(\frac{2d}{\delta_2}\right)} + 2\lambda_2\right)\frac{2\sqrt{s}}{\kappa + 2\lambda_2}\ .$$

*Furthermore, if $\kappa < 3\sqrt{(2/n)\log(2d/\delta_2)}$ and $\lambda_2 = 3\sqrt{(2/n)\log(2d/\delta_2)} - \kappa$, the bound (9) implies*

$$\left\|\hat{\theta}_{\lambda_1,\lambda_2} - \theta_*\right\|_{U_n + \lambda_2 I_d} \leq 2\sqrt{2s}\left(6\sqrt{\frac{2}{n}\log\left(\frac{2d}{\delta_2}\right)} - \kappa\right)^{\frac{1}{2}}$$

$$\leq 4\sqrt[4]{\frac{18s^2}{n}\log\left(\frac{2d}{\delta_2}\right)}\ .$$

### 3.3 Dantzig Selector

Before presenting our fourth main result, we introduce two more conditions on symmetric matrices which underlie standard assumptions in the statistics and compressed sensing literature (Bandeira et al., 2013):

**Definition 2** (Restricted isometry property). *Fix $\epsilon \geq 0$ and $p \in \mathbb{N}$. A symmetric matrix $A \in \mathbb{R}^{d \times d}$ satisfies the $(\epsilon, p)$-restricted isometry property iff*

$$\forall v \in \mathbb{R}^d,\ \|v\|_0 \leq p \implies (1 - \epsilon)\|v\|^2 \leq v^\top A v \leq (1 + \epsilon)\|v\|^2\ .$$

The restricted isometry property states that the curvature of the quadratic form represented by $A$ is bounded around 1 along directions residing in sparse axis-aligned subspaces, or alternatively, that the linear operator represented by $A$ is approximately scale-preserving when applied to sparse inputs. A weaker variant of this property bears resemblance to the Cauchy-Schwarz inequality:

**Definition 3** (Restricted orthogonality property). *Fix $\zeta \geq 0$, $p \in \mathbb{N}$, and $q \in \mathbb{N}$. A symmetric matrix $A \in \mathbb{R}^{d \times d}$ satisfies the $(\zeta, p, q)$-restricted orthogonality property iff*

$$\forall u, v \in \mathbb{R}^d,\ \|u\|_0 \leq p \wedge \|v\|_0 \leq q \wedge \operatorname{supp}(u) \cap \operatorname{supp}(v) = \emptyset \implies \left|u^\top A v\right| \leq \zeta\|u\|\|v\|\ .$$

Under the restricted orthogonality property, $A$ maps sparse vectors with disjoint support to dissimilar outputs. Our fourth main result is a high-probability upper bound on the error of the Dantzig selector in the fixed design setting, under assumptions on $U_n$ similar to the prior literature (Candes & Tao, 2007):

**Theorem 5** (Dantzig selector fixed design upper bound). *Fix $\delta \in (0, 1)$, $\epsilon \in [0, 1)$, and $\zeta \in [0, 1 - \epsilon)$. Assume the samples $(x, y)^{1:n}$ are generated according to the fixed design setting. Assume $U_n$ satisfies the $(\epsilon, 2s)$-restricted isometry property and the $(\zeta, s, 2s)$-restricted orthogonality property. Let $\bar{\theta}_\lambda$ be the Dantzig selector (5) with regularization hyperparameter $\lambda = \sqrt{(2/n)\log(2d/\delta)}$. Then with probability at least $1 - \delta$, the estimation error satisfies the bound*

$$\left\|\bar{\theta}_\lambda - \theta_*\right\| \leq \frac{4}{1 - \epsilon - \zeta}\sqrt{\frac{2s}{n}\log\left(\frac{2d}{\delta}\right)}\ .$$

We include the Dantzig selector in our investigation of sparse contextual CDF regression as an example of variable selection formulated as a linear programming problem (6), as opposed to the usual quadratic programming perspective of lasso estimation. We remark that the $(\epsilon, 3p)$-restricted isometry property implies the $(\epsilon, p, 2p)$-restricted orthogonality property, by Candes & Tao (2005, Lemma 1.1). Hence, Theorem 5 also holds when the sole assumption placed on $U_n$ is the $(\epsilon, 3s)$-restricted isometry property.

We defer the technical details of our proof to Appendix D and provide a high-level summary below. Using a result from the prior statistical literature, we upper-bound the estimation error $\|\Delta\|$ in terms of the components $\|[\Delta]_{S_\dagger}\|$ and $\|[\Delta]_{S_*^c}\|_1$, where $S_\dagger$ is a superset of $S_*$ (43). Utilizing the sparsity of $\theta_*$, we bound the latter component in terms of the former (44). Using another prior result, we upper-bound $\|[\Delta]_{S_\dagger}\|$ in terms of $\|[U_n]_{\langle S_\dagger\rangle}\Delta\|$ (46), which in turn depends on two quantities (48):

- The scalar projections of the sampling errors $\mathrm{I}_{y^{(j)}} - \theta_*^\top \Phi_j$ onto the $d$ basis functions $\Phi_j$, which we upper-bound using similar probabilistic arguments as the proof of Theorem 1.

- The scalar projections of $\mathrm{I}_{y^{(j)}} - \bar{\theta}_\lambda^\top \Phi_j$ onto $\Phi_j$, which we upper-bound using the constraint in (6).

Combining these bounds and applying various norm equivalences completes the proof. The restricted isometry and restricted orthogonality assumptions on $U_n$ are preconditions for the prior results we use, but none of our subsequent arguments depend on these assumptions.

As previously done for the lasso and elastic net estimators, we use a matrix version of Hoeffding's inequality to derive the following high-probability upper bound on the error of the Dantzig selector in the random design setting, which we prove in Appendix D:

**Theorem 6** (Dantzig selector random design upper bound)**.** *Fix $\delta_1 \in (0,1)$, $\delta_2 \in (0, 1-\delta_1)$, $\epsilon \in (0, 1/2)$, and $\zeta \in (0, 1/2 - \epsilon)$. Assume the samples $(x,y)^{1:n}$ are generated according to the random design setting. Assume that for any $n \in \mathbb{N}$, the matrix $\Sigma_n$ satisfies the $(\epsilon, 2s)$-restricted isometry property and the $(\zeta, s, 2s)$-restricted orthogonality property. Assume the sample size is at least $n \geq (8d^2/\min\{\epsilon, \zeta\}^2) \log(2d/\delta_1)$. Let $\bar{\theta}_\lambda$ be the Dantzig selector (5) with regularization hyperparameter $\lambda = \sqrt{(2/n) \log(2d/\delta_2)}$. Then with probability at least $1 - \delta_1 - \delta_2$, the estimation error satisfies the bound*

$$\left\| \bar{\theta}_\lambda - \theta_* \right\| \leq \frac{4}{1 - 2\epsilon - 2\zeta} \sqrt{\frac{2s}{n} \log\left(\frac{2d}{\delta_2}\right)}.$$

### 3.4 Lower Bound

Our last main result is a lower bound on the *minimax $\ell^2$-risk* of sparse contextual CDF regression. To begin this discussion, we introduce some relevant notation. For any $d \in \mathbb{N}$, let $\mathcal{B}_d$ be the universe of all $d$-dimensional bases of contextual CDFs, i.e.,

$$\mathcal{B}_d = \left\{ \Phi : \mathcal{X} \times \mathbb{R} \to [0,1]^d : \forall i \in [d], \forall x \in \mathcal{X}, [\Phi(x, \cdot)]_i \text{ is a CDF} \right\}.$$

For any $x \in \mathcal{X}$, $\theta \in \Delta^{d-1}$, and $\Phi \in \mathcal{B}_d$, let $P_{Y|x,\theta}^\Phi$ denote the probability distribution corresponding to the CDF $\theta^\top \Phi(x, \cdot)$. Given context variables $x^{1:n} \in \mathcal{X}^n$, let $\mathcal{P}_{x^{1:n}}^d$ denote the family of product distributions of $Y^{1:n}$ which are convex combinations of $d$ contextual CDFs, i.e.,

$$\mathcal{P}_{x^{1:n}}^d = \left\{ \bigotimes_{j=1}^n P_{Y|x^{(j)},\theta}^\Phi : \theta \in \Delta^{d-1} \wedge \Phi \in \mathcal{B}_d \right\}.$$

For any $s \leq d$, let $\mathcal{P}_{x^{1:n}}^{d,s} \subset \mathcal{P}_{x^{1:n}}^d$ denote the family of product distributions of $Y^{1:n}$ which are convex combinations of $s$ contextual CDFs chosen from a $d$-dimensional basis, i.e.,

$$\mathcal{P}_{x^{1:n}}^{d,s} = \left\{ \bigotimes_{j=1}^n P_{Y|x^{(j)},\theta}^\Phi : \theta \in \Delta^{d-1} \wedge \|\theta\|_0 = s \wedge \Phi \in \mathcal{B}_d \right\}.$$

Given a distribution $P \in \mathcal{P}_{x^{1:n}}^d$, let $\theta(P) \in \Delta^{d-1}$ denote its parameter and let $S_*(P) = \operatorname{supp}(\theta(P)) \subseteq [d]$. Consequently, $|S_*(P)| = s$ for any $P \in \mathcal{P}_{x^{1:n}}^{d,s}$. For any $n \in \mathbb{N}$, let $\hat{\Theta}_{d,n}$ be the universe of all (possibly randomized) estimators $\hat{\theta} : \mathbb{R}^n \to \mathbb{R}^d$.

Our main result establishes a lower bound on the estimation error of any estimator for sparse contextual CDF regression, and is obtained as a corollary of the lower bound for general contextual CDF regression in Zhang et al. (2024, Theorem 8):

**Corollary 1** (Lower bound)**.** *Fix $d \in \mathbb{N}$, $s \leq d$, and any sufficiently large $n \geq s/2$. Fix $x^{1:n} \in \mathcal{X}^n$. Then, the minimax $\ell^2$-risk of sparse contextual CDF regression satisfies the bound*

$$\mathfrak{R}\left(\theta\left(\mathcal{P}_{x^{1:n}}^{d,s}\right)\right) = \inf_{\hat{\theta} \in \hat{\Theta}_{d,n}} \sup_{P \in \mathcal{P}_{x^{1:n}}^{d,s}} \mathbb{E}_{Y^{1:n} \sim P}\left[\left\| \hat{\theta}(Y^{1:n}) - \theta(P) \right\|\right] = \Omega\left(\sqrt{\frac{s}{n}}\right).$$

We defer the technical details of our proof to Appendix E. In a nutshell, it suffices to consider only the estimation error component corresponding to the indices in $S_*(P)$, thereby reducing the problem to general $s$-dimensional CDF regression. Furthermore, we obtain minimax upper bounds from the high-probability upper bounds in Theorems 1 and 5 by the arguments from Zhang et al. (2024, p. 12). Hence, Corollary 1 establishes the minimax optimality of the lasso and Dantzig selector estimators up to logarithmic factors.

### 3.5 Comparison to Finite-Dimensional Setting

We conclude Section 3 with a summary of the technical differences between the present CDF regression setting and the well-studied finite-dimensional regression setting. Firstly, known upper bounds on estimation error for finite-dimensional sparse regression methods (cf. Wainwright (2019, Equation 7.26)) typically depend on the scales of the noise and features, and assume the sampling noise is a sub-Gaussian random variable. However, the sampling noise $\mathrm{I}_{Y^{(j)}} - \theta_*^\top \Phi_j$ in CDF regression is a random function, which presents challenges when attempting to define its distribution. To circumvent this issue, we analyze the effect of sampling error by considering instead the inner product (defined as an integral over the support $S$) between the sampling noise and the feature CDFs $\Phi_j$. We incorporate this inner product in our formulation of the Dantzig selector (6), and we refer the reader to Equations (13) and (18) in the proof of Theorem 1, Equations (29) and (33) in the proof of Theorem 3, and Equations (40) and (47) in the proof of Theorem 5 for examples of technical steps in our derivations which utilize the inner product between sampling noise and feature CDFs. Ultimately, our proposed estimators admit error bounds independent of the response and noise scales in both the fixed and random design settings (Theorems 1 to 6).

Secondly, the Gramian matrix in finite-dimensional regression is given by the matrix product between the design matrix and its transpose, equivalent to taking the dot product between each pair of training instances. In CDF regression, we instead define the empirical Gramian matrix $U_n$ (1) and expected Gramian matrix $\Sigma_n$ (2) in terms of an integral over the support $S$ of the underlying random variable of the contextual CDFs, corresponding to the inner product between functions in the Hilbert space $\mathcal{L}^2(S, \mathfrak{m})$ as opposed to the dot product between vectors in ordinary $d$-dimensional Euclidean space. In a similar vein, the aforementioned inner product between the sampling noise and feature CDFs has support $[-1, 1]$ due to the boundedness of CDFs (11), enabling us to derive a concentration result on the inner product over $n$ samples using Hoeffding's inequality (12) in Lemma 1.

Thirdly, our analysis of random design settings in CDF regression differs from similar work in the finite-dimensional regime by employing a matrix version of Hoeffding's inequality (Lemma 3) to establish preconditions on the conditioning of $U_n$ given similar assumptions on $\Sigma_n$ and sufficiently many random samples. To justify this step, we bound the operator norm of the empirical Gramian error by using properties of CDFs (26), whereas additional assumptions on the design matrix are required in the analogous finite-dimensional case. We refer readers to (28) in the proof of Theorem 2, (38) in the proof of Theorem 4, and (51) in the proof of Theorem 6 for specific details on the usage of this concentration result in our derivations. We note that we impose the restricted eigenvalue, restricted isometry, and restricted orthogonality assumptions directly on the Gramian matrices $U_n$ and $\Sigma_n$, although such conditions are typically applied to the design matrix in the finite-dimensional setting.

Finally, the boundedness of CDFs plays a role in our derivation of the lower bound on minimax $\ell^2$-risk. In a nutshell, the minimax $\ell^2$-risk scales as the inverse square root of the empirical Gramian matrix's smallest eigenvalue, which in turn is upper-bounded by the Gramian entries and hence the integral of CDF products by (1). For specific details, we refer readers to (54) in the proof of Corollary 1.

Notwithstanding the technical details discussed above, we note that our derivations of the lasso and elastic net error bounds (Theorems 1 and 3) share structural similarities with classical arguments for the corresponding finite-dimensional settings, in that we utilize the optimality of the estimator $\hat{\theta}_\lambda$ to derive a bound relating $\hat{\theta}_\lambda$ and $\theta_*$ by rearranging the empirical risk objective. We refer readers to Wainwright (2019, p. 212-213) for an example of this general technique applied to finite-dimensional regression.

## 4 Proof of Lasso Fixed Design Upper Bound

In this section, we prove Theorem 1. We begin by deriving a concentration bound on the inner products between the sampling errors $I_{y^{(j)}} - \theta_*^\top \Phi_j$ and the basis functions $\Phi_j$, as a consequence of Hoeffding's inequality and the union bound:

**Lemma 1** (Concentration bound). *Fix $\delta \in (0, 1)$. With probability at least $1 - \delta$, it holds that*

$$\left\| \frac{1}{n} \sum_{j=1}^{n} \left\langle I_{y^{(j)}} - \theta_*^\top \Phi_j, \Phi_j \right\rangle \right\|_\infty \leq \sqrt{\frac{2}{n} \log\left(\frac{2d}{\delta}\right)}. \tag{10}$$

*Proof of Lemma 1.* For each $j \in [n]$ and $i \in [d]$, define a random variable

$$Z_{i,j} = \left\langle I_{Y^{(j)}} - \theta_*^\top \Phi_j, [\Phi_j]_i \right\rangle.$$

Let $\mathcal{F}_j = \sigma(Y^{1:j-1})$ denote the $\sigma$-algebra generated by the random variables $Y^{1:j-1}$.[1] The mean of $Z_{i,j}$ is

$$\begin{aligned}
\mathbb{E}[Z_{i,j}] &\overset{(a)}{=} \mathbb{E}[\mathbb{E}[Z_{i,j} \mid \mathcal{F}_j]] \\
&\overset{(b)}{=} \mathbb{E}\left[\mathbb{E}\left[\left\langle I_{Y^{(j)}} - \theta_*^\top \Phi_j, [\Phi_j]_i \right\rangle \mid \mathcal{F}_j\right]\right] \\
&\overset{(c)}{=} \mathbb{E}\left[\left\langle \mathbb{E}\left[I_{Y^{(j)}} - \theta_*^\top \Phi_j \mid \mathcal{F}_j\right], [\Phi_j]_i \right\rangle\right] \\
&\overset{(d)}{=} \mathbb{E}\left[\left\langle 0, [\Phi_j]_i \right\rangle\right] \\
&= 0,
\end{aligned}$$

where (a) holds by the tower rule of expectation, (b) holds by definition of $Z_{i,j}$, (c) holds by Fubini's theorem, and (d) holds because $\mathbb{E}[I_{Y^{(j)}} \mid \mathcal{F}_j] = \theta_*^\top \Phi_j$. Furthermore, the support of $Z_{i,j}$ is $[0, 1]$ because

$$\begin{aligned}
|Z_{i,j}| &\overset{(a)}{=} \left| \int_S \left(I_{y^{(j)}}(t) - \theta_*^\top \Phi_j(t)\right) \phi_i\left(x^{(j)}, t\right) d\mathfrak{m} \right| \\
&\overset{(b)}{\leq} \int_S \left|I_{y^{(j)}}(t) - \theta_*^\top \Phi_j(t)\right| \left|\phi_i\left(x^{(j)}, t\right)\right| d\mathfrak{m} \\
&\overset{(c)}{\leq} \mathfrak{m}(S) \\
&\overset{(d)}{=} 1,
\end{aligned} \tag{11}$$

where (a) holds by definition of inner product, (b) holds by the triangle inequality, (c) holds because CDFs are bounded between 0 and 1, and (d) holds because $\mathfrak{m}$ is a probability measure. Thus, for any $\tau > 0$,

$$\begin{aligned}
\mathbb{P}\left( \left\| \frac{1}{n} \sum_{j=1}^{n} \left\langle I_{Y^{(j)}} - \theta_*^\top \Phi_j, \Phi_j \right\rangle \right\|_\infty \geq \tau \right) &\overset{(a)}{\leq} \sum_{i=1}^{d} \mathbb{P}\left( \left| \frac{1}{n} \sum_{j=1}^{n} Z_{i,j} \right| \geq \tau \right) \\
&\overset{(b)}{\leq} 2d \exp\left(-\frac{n\tau^2}{2}\right),
\end{aligned} \tag{12}$$

where (a) holds by definition of $Z_{i,j}$ and the union bound, and (b) holds by Hoeffding's inequality. Choosing $\tau = \sqrt{2/n \log(2d/\delta)}$ and rearranging, we get $\delta = 2d \exp(-n\tau^2/2)$, and thus

$$\mathbb{P}\left( \left\| \frac{1}{n} \sum_{j=1}^{n} \left\langle I_{Y^{(j)}} - \theta_*^\top \Phi_j, \Phi_j \right\rangle \right\|_\infty \leq \sqrt{\frac{2}{n} \log\left(\frac{2d}{\delta}\right)} \right) \geq 1 - \delta$$

as desired. ∎

---

[1]This lemma also applies in the random design setting by taking $\mathcal{F}_j = \sigma(X^{1:j}, Y^{1:j-1})$.

Now, we are ready to prove Theorem 1.

*Proof of Theorem 1.* Throughout this proof, we restrict to the subset of the probability space where

$$\left\| \frac{1}{n} \sum_{j=1}^{n} \left\langle \mathrm{I}_{y^{(j)}} - \theta_*^\top \Phi_j, \Phi_j \right\rangle \right\|_\infty \leq \sqrt{\frac{2}{n} \log\left(\frac{2d}{\delta}\right)} = \frac{\lambda}{4}, \tag{13}$$

which holds with probability at least $1 - \delta$ by Lemma 1. For notational simplicity, let $\Delta = \hat{\theta}_\lambda - \theta_*$. We follow the argument from Wainwright (2019, p. 212-213). We have

$$
\begin{aligned}
\|\Delta\|_{U_n}^2 &\overset{(a)}{=} \Delta^\top U_n \Delta \\
&\overset{(b)}{=} \Delta^\top \left( \frac{1}{n} \sum_{j=1}^{n} \int_S \Phi_j \Phi_j^\top d\mathfrak{m} \right) \Delta \\
&\overset{(c)}{=} \frac{1}{n} \sum_{j=1}^{n} \int_S \Delta^\top \Phi_j \Phi_j^\top \Delta \, d\mathfrak{m} \\
&\overset{(d)}{=} \frac{1}{n} \sum_{j=1}^{n} \left\| \Delta^\top \Phi_j \right\|_{\mathcal{L}^2(S,\mathfrak{m})}^2 \\
&\overset{(e)}{=} \frac{1}{n} \sum_{j=1}^{n} \left( \left\| \hat{\theta}_\lambda^\top \Phi_j \right\|_{\mathcal{L}^2(S,\mathfrak{m})}^2 + \left\| \theta_*^\top \Phi_j \right\|_{\mathcal{L}^2(S,\mathfrak{m})}^2 \right) - \frac{2}{n} \sum_{j=1}^{n} \left\langle \hat{\theta}_\lambda^\top \Phi_j, \theta_*^\top \Phi_j \right\rangle \\
&\overset{(f)}{=} \frac{1}{n} \sum_{j=1}^{n} \left( \left\| \hat{\theta}_\lambda^\top \Phi_j \right\|_{\mathcal{L}^2(S,\mathfrak{m})}^2 - \left\| \theta_*^\top \Phi_j \right\|_{\mathcal{L}^2(S,\mathfrak{m})}^2 \right) - \frac{2}{n} \sum_{j=1}^{n} \left\langle \Delta^\top \Phi_j, \theta_*^\top \Phi_j \right\rangle,
\end{aligned}
\tag{14}
$$

where (a) holds by definition of weighted $\ell^2$-norm induced by $U_n$, (b) holds by definition of $U_n$, (c) holds by the linearity of integration, (d) holds by definition of inner product between functions, (e) holds by definition of $\Delta$, and (f) holds by definition of $\Delta$ and the linearity of inner product. Next, since $\hat{\theta}_\lambda$ minimizes the objective in (3), it follows that

$$\frac{1}{n} \sum_{j=1}^{n} \left\| \mathrm{I}_{y^{(j)}} - \hat{\theta}_\lambda^\top \Phi_j \right\|_{\mathcal{L}^2(S,\mathfrak{m})}^2 + \lambda \left\| \hat{\theta}_\lambda \right\|_1 \leq \frac{1}{n} \sum_{j=1}^{n} \left\| \mathrm{I}_{y^{(j)}} - \theta_*^\top \Phi_j \right\|_{\mathcal{L}^2(S,\mathfrak{m})}^2 + \lambda \left\| \theta_* \right\|_1. \tag{15}$$

Expanding the squared norms and rearranging, it follows that (cf. Wainwright (2019, Equation 7.29))

$$\frac{1}{n} \sum_{j=1}^{n} \left( \left\| \hat{\theta}_\lambda^\top \Phi_j \right\|_{\mathcal{L}^2(S,\mathfrak{m})}^2 - \left\| \theta_*^\top \Phi_j \right\|_{\mathcal{L}^2(S,\mathfrak{m})}^2 \right) \leq \frac{2}{n} \sum_{j=1}^{n} \left\langle \mathrm{I}_{y^{(j)}}, \Delta^\top \Phi_j \right\rangle + \lambda \left( \left\| \theta_* \right\|_1 - \left\| \hat{\theta}_\lambda \right\|_1 \right). \tag{16}$$

Combining Equations (14) and (16), we obtain

$$\|\Delta\|_{U_n}^2 \leq \underbrace{\frac{2}{n} \sum_{j=1}^{n} \left\langle \mathrm{I}_{y^{(j)}}, \Delta^\top \Phi_j \right\rangle - \frac{2}{n} \sum_{j=1}^{n} \left\langle \Delta^\top \Phi_j, \theta_*^\top \Phi_j \right\rangle}_{①} + \lambda \underbrace{\left( \left\| \theta_* \right\|_1 - \left\| \hat{\theta}_\lambda \right\|_1 \right)}_{②}. \tag{17}$$

Next, we upper-bound ①. We have

$$① \overset{(a)}{=} \Delta^\top \left( \frac{2}{n} \sum_{j=1}^{n} \left\langle \mathrm{I}_{y^{(j)}} - \theta_*^\top \Phi_j, \Phi_j \right\rangle \right) \tag{18}$$

$$\stackrel{(b)}{\leq} \|\Delta\|_1 \left\| \frac{2}{n} \sum_{j=1}^n \left\langle \mathrm{I}_{y^{(j)}} - \theta_*^\top \Phi_j, \Phi_j \right\rangle \right\|_\infty$$

$$\stackrel{(c)}{\leq} \frac{\lambda}{2} \|\Delta\|_1 \, , \tag{19}$$

where (a) holds by the linearity of inner product, (b) follows from Hölder's inequality, and (c) holds by substituting in (13). Next, we upper-bound ②. We have

$$② \stackrel{(a)}{=} \sum_{i=1}^d |[\theta_*]_i| - \sum_{i=1}^d |[\theta_* + \Delta]_i|$$

$$\stackrel{(b)}{=} \sum_{i \in S_*} |[\theta_*]_i| - \sum_{i \in S_*} |[\theta_* + \Delta]_i| - \sum_{i \in S_*^c} |[\Delta]_i|$$

$$\stackrel{(c)}{=} \left\| [\theta_*]_{S_*} \right\|_1 - \left\| [\theta_*]_{S_*} + [\Delta]_{S_*} \right\|_1 - \left\| [\Delta]_{S_*^c} \right\|_1$$

$$\stackrel{(d)}{\leq} \left\| [\Delta]_{S_*} \right\|_1 - \left\| [\Delta]_{S_*^c} \right\|_1 \, , \tag{20}$$

where (a) holds by definition of $\ell^1$-norm and $\Delta$, (b) holds because $\mathrm{supp}(\theta_*) = S_*$, (c) holds by definition of $\ell^1$-norm, and (d) holds by the triangle inequality. Combining Equations (17), (19) and (20), we obtain

$$\|\Delta\|_{U_n}^2 \leq \lambda \left( \frac{3}{2} \left\| [\Delta]_{S_*} \right\|_1 - \frac{1}{2} \left\| [\Delta]_{S_*^c} \right\|_1 \right) \tag{21}$$

$$\leq \frac{3\lambda}{2} \left\| [\Delta]_{S_*} \right\|_1 \tag{22}$$

$$\stackrel{(a)}{\leq} \frac{3\lambda}{2} \sqrt{s} \left\| [\Delta]_{S_*} \right\|$$

$$\leq \frac{3\lambda}{2} \sqrt{s} \, \|\Delta\|$$

$$\stackrel{(b)}{\leq} \frac{3\lambda}{2} \sqrt{\frac{s}{\kappa}} \, \|\Delta\|_{U_n} \, ,$$

where (a) holds by the equivalence between $\ell^1$- and $\ell^2$-norms, and (b) follows from the restricted eigenvalue condition on $U_n$ since rearranging (21) yields $\|[\Delta]_{S_*^c}\|_1 \leq 3\|[\Delta]_{S_*}\|_1$. Thus, the estimation error satisfies the bounds

$$\|\Delta\|_{U_n} \leq \frac{3\lambda}{2} \sqrt{\frac{s}{\kappa}} \stackrel{(a)}{=} 6\sqrt{\frac{2s}{\kappa n} \log\left(\frac{2d}{\delta}\right)} \quad \text{and} \quad \|\Delta\| \stackrel{(b)}{\leq} \frac{1}{\sqrt{\kappa}} \|\Delta\|_{U_n} = \frac{6}{\kappa} \sqrt{\frac{2s}{n} \log\left(\frac{2d}{\delta}\right)} \, ,$$

where (a) holds by substituting in $\lambda$ and (b) follows from the restricted eigenvalue condition on $U_n$. ∎

## 5  Numerical Simulations

In this section, we numerically simulate the data generation processes described in Section 1.2 and empirically evaluate the accuracy of our proposed lasso (3) and elastic net (4) estimators on synthetic data. For mathematical conciseness, we consider a basis of contextual Bernoulli CDFs, which yields a closed-form expression for the induced norm $\|\cdot\|_{\mathcal{L}^2(S,\mathfrak{m})}$ in the training objective. Formally, let $\mathcal{X} = [0,1]^d$ be the space of all $d$-tuples $x = (x_1, \ldots, x_d)$ of Bernoulli parameters, and define the basis functions

$$\forall i \in [d], \ \phi_i(x,t) = \begin{cases} 1 - x_i, & \text{if } 0 \leq t < 1 \, , \\ 1, & \text{if } t = 1 \, , \end{cases}$$

where $\phi_i(x,\cdot)$ is the CDF of a $\mathsf{Bernoulli}(x_i)$ random variable. It follows that $F(x,\cdot) = \theta_*^\top \Phi(x,\cdot)$ is the CDF of a $\mathsf{Bernoulli}(\theta_*^\top x)$ random variable, because

$$F(x,t) \;=\; \sum_{i=1}^d [\theta_*]_i \cdot \begin{cases} 1 - x_i\,, & \text{if } 0 \le t < 1 \\ 1\,, & \text{if } t = 1 \end{cases} \overset{(a)}{=} \begin{cases} 1 - \theta_*^\top x\,, & \text{if } 0 \le t < 1\,, \\ 1\,, & \text{if } t = 1\,, \end{cases}$$

where (a) holds because $\theta_*$ is a PMF. Let $\mathfrak{m}$ be the uniform measure over the support $S = [0,1]$. Then, the log-likelihood term in Equations (3) and (4) is equivalent to the canonical least squares formulation because

$$\sum_{j=1}^n \big\| \mathrm{I}_{y^{(j)}} - \theta^\top \Phi_j \big\|_{\mathcal{L}^2(S,\mathfrak{m})}^2 \overset{(a)}{=} \sum_{j=1}^n \int_0^1 \big( \mathrm{I}_{y^{(j)}}(t) - \theta^\top \Phi_j(t) \big)^2 \, dt$$

$$\overset{(b)}{=} \sum_{j=1}^n \Big( \mathbb{1}\big\{ y^{(j)} = 0 \big\} - \big( 1 - \theta^\top x^{(j)} \big) \Big)^2$$

$$= \sum_{j=1}^n \big( \theta^\top x^{(j)} - y^{(j)} \big)^2$$

$$\overset{(c)}{=} \| A\theta - b \|^2 \,,$$

where (a) follows from the support $S = [0,1]$ and the choice of measure, (b) holds by substituting in the definitions of $\mathrm{I}_{y^{(j)}}$ and $\Phi_j$, and the matrix $A \in \mathbb{R}^{n \times d}$ and vector $b \in \mathbb{R}^n$ in (c) are defined as $[A]_{\langle j \rangle} = x^{(j)\top}$ and $[b]_j = y^{(j)}$. For specific details regarding our implementation, we refer interested readers to our Python code at https://github.com/enchainingrealm/SparseContextualCDFRegression.

In our experiments, we compare the $\ell^2$-norm estimation error of our proposed lasso and elastic net estimators against the ridge regression baseline introduced in Zhang et al. (2024, Section 3.1). For the $\ell^2$-regularization hyperparameter of ridge regression and the $\ell^1$-regularization hyperparameter of lasso and elastic net regression, we use $\lambda = 4\sqrt{(2/n) \log(2d/\delta)}$ as specified in Theorems 1 and 3, with $\delta = 0.001$. For experimental convenience, we use various fixed values of $\lambda_2$ for the elastic net estimator, which we report in Figures 1 and 2. Different values of $\lambda$ and $\delta$ produced qualitatively similar results. We investigate how the estimation errors scale with various problem dimensions, and report means and standard deviations over 30 independent random trials in Figures 1 and 2 for each configuration under consideration. We train all models on the same set of generated samples in each random trial.

## 5.1 Fixed Design

First, we consider the fixed design setting with context variables $x^{(j)} = (x_1^{(j)}, \dots, x_d^{(j)})$ given by

$$x_i^{(j)} = \begin{cases} 1 - 2x_{\mathsf{val}}^{(j)}\,, & \text{if } i \equiv j \pmod d \\ 1 - x_{\mathsf{val}}^{(j)}\,, & \text{if } i \not\equiv j \pmod d \end{cases} \quad \text{with} \quad x_{\mathsf{val}}^{(j)} = \begin{cases} \frac{1}{2}\,, & \text{if } j \le d \\ \frac{\mu_{\min}(M_{j-1})}{\alpha_j}\,, & \text{if } j > d \end{cases}$$

$$\text{and} \quad M_j = \big( 1 - x^{(j)} \big)\big( 1 - x^{(j)} \big)^\top + \frac{1}{n} \sum_{k=1}^{j-1} \big( 1 - x^{(k)} \big)\big( 1 - x^{(k)} \big)^\top \,,$$

where $\alpha_j$ is initialized as $\alpha_{d+1} = \mu_{\min}(M_d)/2$ and is doubled as necessary on each $j$ iteration to ensure $x^{(j)} \in \mathcal{X}$. To investigate the effect of the sample size $n$, we choose 100 logarithmically spaced points for $n$ from $10^4$ to $10^6$, and fix the CDF basis dimension $d = 10$ and parameter sparsity $s = 5$. Figure 1(a) graphs the estimation errors against $n$ on a log-log plot. The lasso trend line has slope $-1/2$, matching the theoretical $O(1/\sqrt{n})$ bound in Theorem 1 and substantially outperforming the ridge baseline. The elastic net estimator achieves results in between the lasso and ridge methods. As a further point of comparison, we repeat this experiment using handpicked regularization hyperparameters for the ridge estimator and plot the results in Figure 1(b). Values of $\lambda$ less than $10^{-3}$ or greater than $10^{-1}$ produced results comparable to the yellow and purple trend lines, respectively. Observe that the accuracy of the ridge estimator with $\lambda = 10^{-3}$ matches the lasso estimator when $n \le 10^5$, but quickly plateaus for larger sample sizes. This control experiment confirms

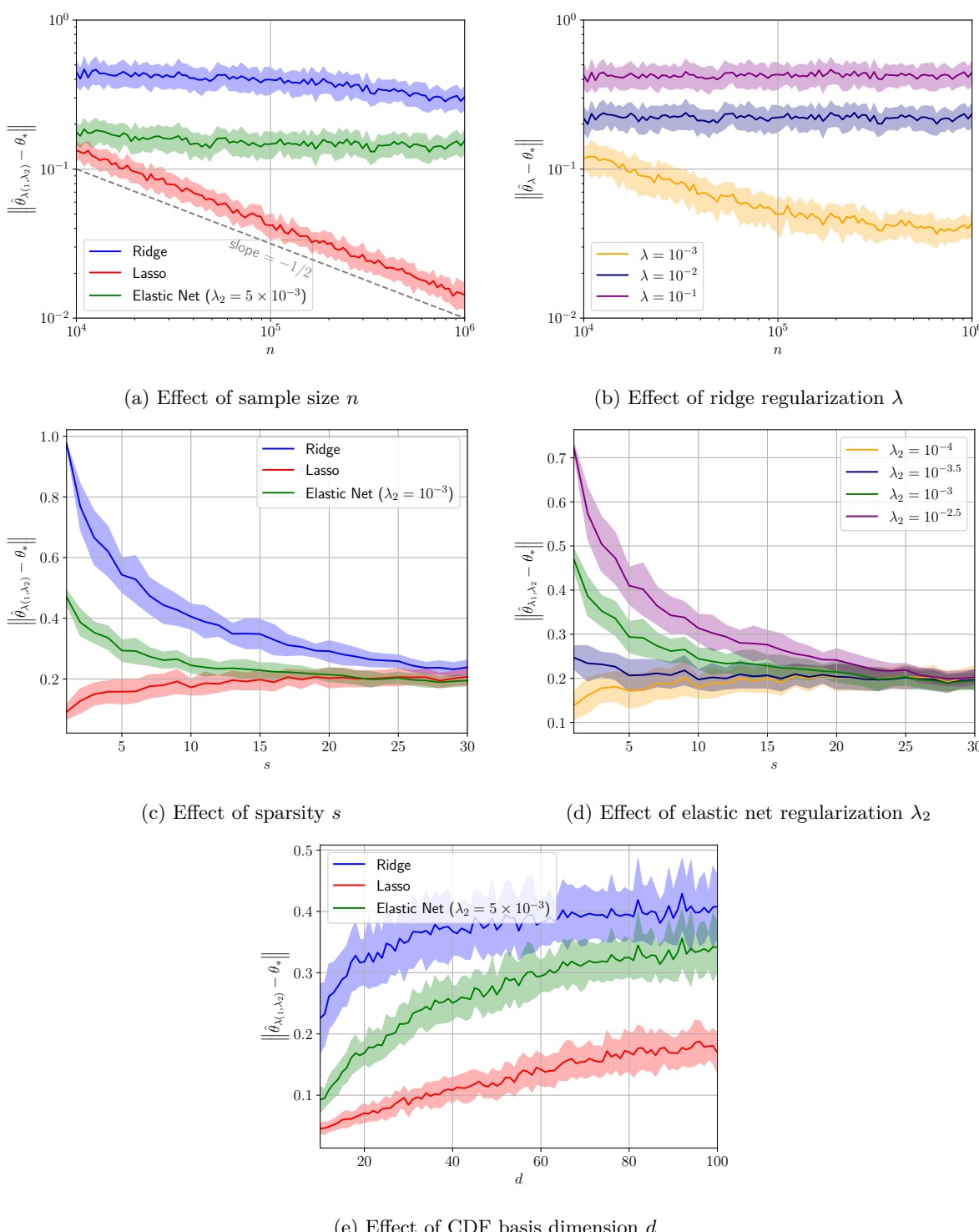

(a) Effect of sample size $n$

(b) Effect of ridge regularization $\lambda$

(c) Effect of sparsity $s$

(d) Effect of elastic net regularization $\lambda_2$

(e) Effect of CDF basis dimension $d$

Figure 1: Means and standard deviations of ridge, lasso, and elastic net estimation errors against various hyperparameters in the synthetic Bernoulli experiments for the fixed design setting. Figures 1(a), 1(c) and 1(e) contrast the ridge, lasso, and elastic net estimators with common regularization hyperparameter $\lambda = \lambda_1 = 4\sqrt{(2/n)\log(2d/\delta)}$, as defined in Theorems 1 and 3. For further comparison, Figure 1(b) plots the ridge estimation error against $n$ for various values of $\lambda$, and Figure 1(d) plots the elastic net estimation error against $s$ for various values of $\lambda_2$.

that the superior accuracy of our lasso estimator in the large $n$ setting cannot be emulated by the ridge baseline regardless of hyperparameter tuning.

Secondly, we investigate how the estimation errors scale with the sparsity $s$ of the true parameter $\theta_*$. We fix the sample size $n = 10^5$ and CDF basis dimension $d = 30$, and consider $s$ from 1 to 30. Figure 1(c) graphs the estimation errors against $s$ on a linear plot. The lasso trend line reflects the theoretical $O(\sqrt{s})$ bound in Theorem 1, and our lasso estimator notably outperforms the ridge baseline in the sparse regime $s \ll d$. To interpret the decreasing ridge trend line, note that $\|\theta_*\|_1 = 1$ for any value of $s$ (because $\theta_*$ is a PMF), and so denser parameter vectors (with greater $s$) tend to have smaller $\|\theta_*\|$. Thus, the ridge regularization penalty $\lambda\|\theta_*\|^2$ at the true parameter vector decreases as $s$ increases, leading to improved accuracy. For additional comparison, we repeat this experiment for the elastic net estimator using handpicked values of $\lambda_2$ and plot the results in Figure 1(d). As expected, the elastic net estimation error interpolates continuously between the lasso and ridge estimation errors as $\lambda_2$ increases. Values of $\lambda_2$ outside the range visualized in Figure 1(d) produced similar results.

Lastly, we investigate the dependence of the estimation errors on the CDF basis dimension $d$. We fix the sample size $n = 10^5$ and parameter sparsity $s = 10$, and consider $d$ from 10 to 100. Figure 1(e) graphs the estimation errors against $d$ on a linear plot. The ridge and lasso trend lines indicate the respective theoretical bounds of $O(\sqrt{d})$ (Zhang et al., 2024, Section 3.2) and $O(\sqrt{\log d})$ (Theorem 1).

### 5.2 Random Design

Next, we consider the random design setting with context vectors $x^{(j)} = (x_1^{(j)}, \ldots, x_d^{(j)})$ sampled i.i.d. from a $\mathsf{Uniform}(0,1)^d$ distribution. We investigate the effect of the sample size $n$, sparsity $s$, and CDF basis dimension $d$ on the ridge, lasso, and elastic net estimation errors, as was done for the fixed design setting above, and plot the results in Figures 2(a) to 2(c) respectively. The qualitative trends in the random design experiments are comparable to the corresponding fixed design experiments, although the variance among the 30 trials for each configuration is lower in the random design setting. We conjecture that the variance is reduced by virtue of the context vectors in the random design experiments being more typical instances of the problem, as opposed to the fixed design setting where the context vectors were specifically chosen to be hard instances which may amplify the effect of the randomness in $\theta_*$. On a different note, we remark that the elastic net error asymptotes for $n \approx 10^6$ in Figure 2(a) because $\lambda_2$ is fixed for experimental convenience and does not decrease with $n$, although a more refined hyperparameter setting could avoid this effect in principle. Lastly, we compare different values of $\lambda_2$ for the elastic net estimator in Figure 2(d) and observe the expected interpolation between the ridge and lasso trendlines as $d$ ranges from 10 to 100.

## 6 Conclusion

In this paper, we introduced the task of sparse contextual CDF regression and proposed three basis selection techniques for this problem stemming from the canonical lasso, elastic net, and Dantzig selector regression methods. We derived upper bounds of $\tilde{O}(\sqrt{s/n})$ and $\tilde{O}(\sqrt{s}/\sqrt[4]{n})$ on estimation error, and obtained a matching lower bound on minimax risk to establish the optimality of our proposed lasso and Dantzig selector estimators. In particular, our estimation bounds have sub-polynomial dependence on the dimension $d$ of the CDF regression basis, enabling our methods to perform basis selection with exponentially many irrelevant features and furthering progress towards the ultimate goal of general contextual CDF estimation.

We suggest three directions for future work. Firstly, our present analysis holds only when $d$ is finite. A natural continuation of our research may investigate similar basis selection methods for CDF regression with infinite-dimensional feature maps. Another promising follow-up direction is to generalize our results to CDF regression with the least absolute deviation ($\ell^1$-) loss, in the vein of previous work which combines robust regression and variable selection (Wang et al., 2007). Lastly, prior literature has established looser estimation error bounds on finite-dimensional lasso regression without the restricted isometry property (Zhao & Yu, 2006; Meinshausen & Yu, 2009). In this spirit, future work may aim to determine unified and weaker necessary conditions for analyzing the functional lasso, elastic net, and Dantzig estimators proposed in this paper.

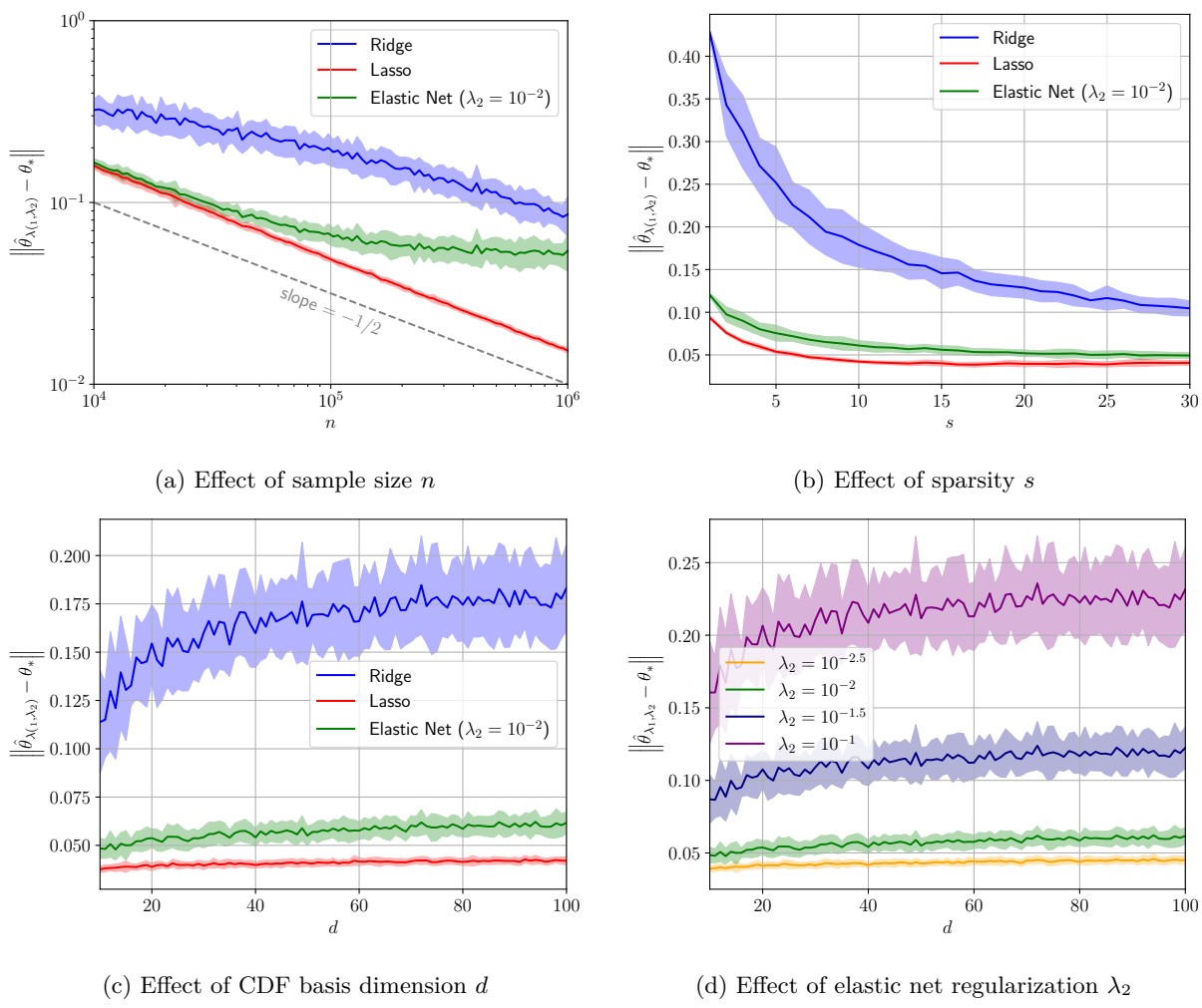

(a) Effect of sample size $n$

(b) Effect of sparsity $s$

(c) Effect of CDF basis dimension $d$

(d) Effect of elastic net regularization $\lambda_2$

Figure 2: Means and standard deviations of ridge, lasso, and elastic net estimators against various hyperparameters in the synthetic Bernoulli experiments for the random design setting. Figures 2(a) to 2(c) contrast the ridge, lasso, and elastic net estimators with common regularization hyperparameter $\lambda = \lambda_1 = 4\sqrt{(2/n)\log(2d/\delta)}$, as defined in Theorems 1 and 3. For further comparison, Figure 2(d) plots the elastic net estimation error against $d$ for various values of $\lambda_2$.

Overall, our main contributions and proposed future directions indicate that contextual CDF estimation remains a fruitful area of theoretical investigation, accompanied by immediate relevance to a profusion of downstream scientific applications.

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

## A   Notation

Let $\mathbb{N}$ denote the natural numbers starting from 1. Let $[n] = \mathbb{Z} \cap [1, n]$ denote the set of natural numbers from 1 to $n$. Let $\mathbb{1}\{\cdot\}$ denote the Iverson bracket. When discussing empirical CDFs, we use $\mathrm{I}_a(t) = \mathbb{1}\{t \geq a\}$ to denote translated unit step functions. Let $I_d$ denote the $d \times d$ identity matrix. Let $\Delta^{d-1} \subset \mathbb{R}^d$ denote the $(d-1)$-dimensional probability simplex. Let $\arg_k \max_{x \in S} f(x)$ denote the top-$k$ maximizers of a function $f$ over a set $S$. In the context of Landau notation, let $\tilde{O}(\cdot)$ denote asymptotic upper bounds with hidden logarithmic or sub-logarithmic factors. Throughout this paper, all logarithms have base $e$ and all vectors are column vectors. When invoking a theorem, we use the notation $x := y$ to pass value $y$ to variable $x$ in the theorem statement.

Given a vector $x \in \mathbb{R}^d$, let $[x]_i$ denote its $i$th entry. For any set $S \subseteq [d]$, let $[x]_S \in \mathbb{R}^{|S|}$ denote the entries of $x$ corresponding to the indices in $S$. In this context, let $S^c = [d] - S$ denote set complement with respect to the universe of indices. Let $\mathrm{supp}(x)$ denote the support of $x$ and let $\|x\|_0 = |\mathrm{supp}(x)|$ denote the number of non-zero entries in $x$. Let $\|x\|_p$ denote the $\ell^p$-norm of $x$ for any $p \in [1, \infty]$, let $\|x\|$ denote its Euclidean ($\ell^2$-) norm, and let $\|x\|_A = \sqrt{x^\top A x}$ denote its weighted $\ell^2$-norm induced by a positive definite matrix $A \in \mathbb{R}^{d \times d}$.

Given a matrix $A \in \mathbb{R}^{m \times n}$, let $[A]_{i,j}$ denote its entry at row $i$ and column $j$. For any set $S \subseteq [m]$, let $[A]_{\langle S \rangle} \in \mathbb{R}^{|S| \times n}$ denote the submatrix obtained from $A$ by extracting the rows corresponding to the indices in $S$. For any sets $S \subseteq [m]$ and $T \subseteq [n]$, let $[A]_{S,T} \in \mathbb{R}^{|S| \times |T|}$ denote the submatrix obtained from $A$ by extracting the rows corresponding to the indices in $S$ and the columns corresponding to the indices in $T$. Let $\|A\|$ denote the induced $\ell^2$-operator norm of $A$, let $\mu_{\min}(A)$ denote its minimum eigenvalue, let $\mu_{\max}(A)$ denote its maximum eigenvalue, and let $\sigma_{\min}(A)$ denote its minimum singular value.

We denote random variables with uppercase letters and realizations with lowercase letters. We use the notation $\mathbb{E}_{X \sim \mathcal{P}}[\cdot]$ to emphasize the random variable and distribution which an expectation is taken over. Given probability distributions $P_1, \ldots, P_n$ over random variables $X_1, \ldots, X_n$ respectively, let $\bigotimes_{j=1}^n P_j$ be the product distribution over $(X_1, \ldots, X_n)$. Given a probability measure $\mathfrak{m}$ on a set $S$, define an inner product between functions $f, g : S \to \mathbb{R}$ as $\langle f, g \rangle = \int_S f(t)\, g(t)\, d\mathfrak{m}$, and denote the norm induced by this inner product as $\|f\|_{\mathcal{L}^2(S, \mathfrak{m})} = \sqrt{\langle f, f \rangle}$. Throughout this paper, integrals and inner products are defined entry-wise for vector-valued and matrix-valued functions.

## B   Proof of Lasso Random Design Upper Bound

In this section, we prove Theorem 2. We begin by establishing a restricted eigenvalue condition on the empirical Gramian matrix $U_n$ with high probability:

**Lemma 2** (Empirical Gramian restricted eigenvalue condition). *Fix $\delta \in (0, 1)$. Assume that for any $n \in \mathbb{N}$, the matrix $\Sigma_n$ satisfies the $(\kappa, \gamma)$-restricted eigenvalue condition over $S_*$. Then, for any $n \geq (32d^2/\kappa^2) \log(d/\delta)$, with probability at least $1 - \delta$, the matrix $U_n$ satisfies the $(\kappa/2, \gamma)$-restricted eigenvalue condition over $S_*$.*

*Proof of Lemma 2.* Fix $n \geq (32d^2/\kappa^2) \log(d/\delta)$ and $v \in C_\gamma(S_*)$. We have

$$
\begin{aligned}
v^\top U_n v &= v^\top \Sigma_n v + v^\top (U_n - \Sigma_n) v \\
&\geq v^\top \Sigma_n v + \mu_{\min}(U_n - \Sigma_n) \|v\|^2 \\
&\overset{(a)}{\geq} \kappa \|v\|^2 + \mu_{\min}(U_n - \Sigma_n) \|v\|^2 ,
\end{aligned}
$$

where (a) holds by the restricted eigenvalue condition on $\Sigma_n$. For each $j \in [n]$, define a matrix

$$
D_j = \int_S \Phi_j \Phi_j^\top d\mathfrak{m} - \mathbb{E}\left[ \int_S \Phi_j \Phi_j^\top d\mathfrak{m} \right].
$$

Observe that $U_n - \Sigma_n = (1/n) \sum_{j=1}^n D_j$. Our proof uses the following result which we distill from the random matrix theory literature and restate for convenience:

**Lemma 3** (Matrix Hoeffding, [Tropp 2012], Theorem 1.3). *Let $A_1, \ldots, A_n \in \mathbb{R}^{d \times d}$ be independent random symmetric matrices, each with mean $\mathbb{E}[A_k] = 0$ and operator norm $\|A_k\| \le b_k$. Then, for any $\tau > 0$,*

$$\mathbb{P}\left(\mu_{\max}\left(\sum_{k=1}^{n} A_k\right) \ge \tau\right) \le d \exp\left(-\frac{\tau^2}{8 \sum_{k=1}^{n} b_k^2}\right). \tag{23}$$

*Furthermore, two direct consequences of (23) are*

$$\mathbb{P}\left(\mu_{\max}\left(\frac{1}{n}\sum_{k=1}^{n} A_k\right) \ge \tau\right) = \mathbb{P}\left(\mu_{\max}\left(\sum_{k=1}^{n} A_k\right) \ge n\tau\right)$$
$$\overset{(a)}{\le} d \exp\left(-\frac{n^2 \tau^2}{8 \sum_{k=1}^{n} b_k^2}\right), \tag{24}$$

*where (a) holds by applying (23), and*

$$\mathbb{P}\left(\mu_{\min}\left(\frac{1}{n}\sum_{k=1}^{n} A_k\right) \le -\tau\right) = \mathbb{P}\left(-\mu_{\min}\left(\sum_{k=1}^{n} A_k\right) \ge n\tau\right)$$
$$= \mathbb{P}\left(\mu_{\max}\left(\sum_{k=1}^{n} -A_k\right) \ge n\tau\right)$$
$$\overset{(b)}{\le} d \exp\left(-\frac{n^2 \tau^2}{8 \sum_{k=1}^{n} b_k^2}\right), \tag{25}$$

*where (b) holds by (24) because $\mathbb{E}[-A_k] = -\mathbb{E}[A_k] = 0$ and $\|-A_k\| = \|A_k\| \le b_k$.*

By inspection, the mean of $D_j$ is $\mathbb{E}[D_j] = 0$. Moreover, the norm of $D_j$ can be upper-bounded as

$$\|D_j\| \overset{(a)}{=} \max_{\|v\|=1} \left|v^\top D_j v\right|$$
$$\overset{(b)}{=} \max_{\|v\|=1} \left|v^\top \left(\int_S \Phi_j \Phi_j^\top \, d\mathfrak{m}\right) v - v^\top \mathbb{E}\left[\int_S \Phi_j \Phi_j^\top \, d\mathfrak{m}\right] v\right|$$
$$= \max_{\|v\|=1} \left|\int_S \left(v^\top \Phi_j\right)^2 d\mathfrak{m} - \mathbb{E}\left[\int_S \left(v^\top \Phi_j\right)^2 d\mathfrak{m}\right]\right|$$
$$\overset{(c)}{\le} \max_{\|v\|=1} \max\left\{\int_S \left(v^\top \Phi_j\right)^2 d\mathfrak{m}, \mathbb{E}\left[\int_S \left(v^\top \Phi_j\right)^2 d\mathfrak{m}\right]\right\}$$
$$\overset{(d)}{\le} \max_{\|v\|=1} \max\left\{\int_S \|v\|^2 \|\Phi_j\|^2 d\mathfrak{m}, \mathbb{E}\left[\int_S \|v\|^2 \|\Phi_j\|^2 d\mathfrak{m}\right]\right\}$$
$$= \max\left\{\int_S \|\Phi_j\|^2 d\mathfrak{m}, \mathbb{E}\left[\int_S \|\Phi_j\|^2 d\mathfrak{m}\right]\right\}$$
$$\overset{(e)}{=} \max\left\{\sum_{i=1}^{d} \int_S \phi_i\left(x^{(j)}, t\right)^2 d\mathfrak{m}, \sum_{i=1}^{d} \mathbb{E}\left[\int_S \phi_i\left(X^{(j)}, t\right)^2 d\mathfrak{m}\right]\right\}$$
$$\overset{(f)}{\le} \sum_{i=1}^{d} \mathfrak{m}(S) \tag{26}$$
$$\overset{(g)}{=} d, \tag{27}$$

where (a) holds because $D_j$ is symmetric, (b) holds by definition of $D_j$, (c) holds because $|\alpha - \beta| \le \max\{\alpha, \beta\}$ for all non-negative numbers $\alpha$ and $\beta$, (d) holds by the Cauchy-Schwarz inequality, (e) holds by definition of $\ell^2$-norm, (f) holds because $\phi_i$ is a contextual CDF for all $i \in [d]$, and (g) holds because $\mathfrak{m}$ is a probability measure. Thus, with probability at least $1 - \delta$, it holds that

$$v^\top U_n v \ge \kappa \|v\|^2 + \mu_{\min}\left(\frac{1}{n}\sum_{j=1}^{n} D_j\right) \|v\|^2$$

$$\overset{(a)}{\geq} \kappa \|v\|^2 - 2d\sqrt{\frac{2}{n} \log\left(\frac{d}{\delta}\right)} \|v\|^2$$

$$\overset{(b)}{\geq} \kappa \|v\|^2 - \frac{\kappa}{2} \|v\|^2$$

$$\overset{(c)}{=} \frac{\kappa}{2} \|v\|^2 ,$$

where (a) follows from (25) after invoking Lemma 3 with $A_k := D_j$ and $\tau := 2d\sqrt{(2/n)\log(d/\delta)}$, and (b) follows because $n \geq (32d^2/\kappa^2)\log(d/\delta)$. Since $v \in C_\gamma(S_*)$ is arbitrary, (c) is the $(\kappa/2, \gamma)$-restricted eigenvalue condition over $S_*$ as desired. ∎

Now, we are ready to prove Theorem 2.

*Proof of Theorem 2.* Let $\mathcal{E}$ be the event that $U_n$ satisfies the $(\kappa/2, 3)$-restricted eigenvalue condition over $S_*$. Invoking Lemma 2 with $\delta := \delta_1$ and $\gamma := 3$, we get

$$\mathbb{P}(\mathcal{E}) \geq 1 - \delta_1 . \tag{28}$$

Next, let $\mathcal{F}$ be the event that the estimation error satisfies the desired bounds

$$\left\|\hat{\theta}_\lambda - \theta_*\right\|_{U_n} \leq 6\sqrt{\frac{2s}{(\kappa/2)n} \log\left(\frac{2d}{\delta_2}\right)} = 12\sqrt{\frac{s}{\kappa n} \log\left(\frac{2d}{\delta_2}\right)},$$

$$\left\|\hat{\theta}_\lambda - \theta_*\right\| \leq \frac{6}{\kappa/2}\sqrt{\frac{2s}{n} \log\left(\frac{2d}{\delta_2}\right)} = \frac{12}{\kappa}\sqrt{\frac{2s}{n} \log\left(\frac{2d}{\delta_2}\right)}.$$

Invoking Theorem 1 with $\kappa := \kappa/2$ and $\delta := \delta_2$, we get $\mathbb{P}(\mathcal{F}|\mathcal{E}) \geq 1 - \delta_2$. Thus,

$$\mathbb{P}(\mathcal{F}) \geq \mathbb{P}(\mathcal{E} \cap \mathcal{F}) = \mathbb{P}(\mathcal{E})\,\mathbb{P}(\mathcal{F} \mid \mathcal{E}) \geq (1 - \delta_1)(1 - \delta_2) \geq 1 - \delta_1 - \delta_2$$

as desired. ∎

## C    Proofs of Elastic Net Upper Bounds

In this section, we prove Theorems 3 and 4.

*Proof of Theorem 3.* Throughout this proof, we restrict to the subset of the probability space where

$$\left\|\frac{1}{n}\sum_{j=1}^n \left\langle I_{y^{(j)}} - \theta_*^\top \Phi_j, \Phi_j \right\rangle\right\|_\infty \leq \sqrt{\frac{2}{n} \log\left(\frac{2d}{\delta}\right)} = \frac{\lambda_1}{4}, \tag{29}$$

which holds with probability at least $1 - \delta$ by Lemma 1. For notational simplicity, let $\Delta = \hat{\theta}_{\lambda_1, \lambda_2} - \theta_*$. We follow the argument from Wainwright (2019, p. 212-213). We have

$$\|\Delta\|_{U_n + \lambda_2 I_d}^2 \overset{(a)}{=} \Delta^\top (U_n + \lambda_2 I_d) \Delta$$

$$\overset{(b)}{=} \Delta^\top \left(\frac{1}{n}\sum_{j=1}^n \int_S \Phi_j \Phi_j^\top d\mathfrak{m}\right) \Delta + \lambda_2 \|\Delta\|^2$$

$$\overset{(c)}{=} \frac{1}{n}\sum_{j=1}^n \int_S \Delta^\top \Phi_j \Phi_j^\top \Delta \, d\mathfrak{m} + \lambda_2 \|\Delta\|^2$$

$$\overset{(d)}{=} \frac{1}{n} \sum_{j=1}^{n} \left\| \Delta^\top \Phi_j \right\|_{\mathcal{L}^2(S,\mathfrak{m})}^2 + \lambda_2 \left\| \Delta \right\|^2$$

$$\overset{(e)}{=} \frac{1}{n} \sum_{j=1}^{n} \left( \left\| \hat{\theta}_{\lambda_1,\lambda_2}^\top \Phi_j \right\|_{\mathcal{L}^2(S,\mathfrak{m})}^2 + \left\| \theta_*^\top \Phi_j \right\|_{\mathcal{L}^2(S,\mathfrak{m})}^2 \right) + \lambda_2 \left( \left\| \hat{\theta}_{\lambda_1,\lambda_2} \right\|^2 + \left\| \theta_* \right\|^2 \right)$$

$$- \frac{2}{n} \sum_{j=1}^{n} \left\langle \hat{\theta}_{\lambda_1,\lambda_2}^\top \Phi_j, \theta_*^\top \Phi_j \right\rangle - 2\lambda_2 \hat{\theta}_{\lambda_1,\lambda_2}^\top \theta_*$$

$$\overset{(f)}{=} \frac{1}{n} \sum_{j=1}^{n} \left( \left\| \hat{\theta}_{\lambda_1,\lambda_2}^\top \Phi_j \right\|_{\mathcal{L}^2(S,\mathfrak{m})}^2 - \left\| \theta_*^\top \Phi_j \right\|_{\mathcal{L}^2(S,\mathfrak{m})}^2 \right) + \lambda_2 \left( \left\| \hat{\theta}_{\lambda_1,\lambda_2} \right\|^2 - \left\| \theta_* \right\|^2 \right)$$

$$- \frac{2}{n} \sum_{j=1}^{n} \left\langle \Delta^\top \Phi_j, \theta_*^\top \Phi_j \right\rangle - 2\lambda_2 \Delta^\top \theta_* , \tag{30}$$

where (a) holds by definition of weighted $\ell^2$-norm induced by $U_n$ (cf. Zou & Hastie (2005)), (b) holds by definition of $U_n$, (c) holds by the linearity of integration, (d) holds by definition of inner product between functions, (e) holds by definition of $\Delta$, and (f) holds by definition of $\Delta$ and the linearity of inner product. Next, since $\hat{\theta}_{\lambda_1,\lambda_2}$ minimizes the objective in (4), it follows that

$$\frac{1}{n} \sum_{j=1}^{n} \left\| \mathrm{I}_{y^{(j)}} - \hat{\theta}_{\lambda_1,\lambda_2}^\top \Phi_j \right\|_{\mathcal{L}^2(S,\mathfrak{m})}^2 + \lambda_1 \left\| \hat{\theta}_{\lambda_1,\lambda_2} \right\|_1 + \lambda_2 \left\| \hat{\theta}_{\lambda_1,\lambda_2} \right\|^2$$

$$\leq \frac{1}{n} \sum_{j=1}^{n} \left\| \mathrm{I}_{y^{(j)}} - \theta_*^\top \Phi_j \right\|_{\mathcal{L}^2(S,\mathfrak{m})}^2 + \lambda_1 \left\| \theta_* \right\|_1 + \lambda_2 \left\| \theta_* \right\|^2 .$$

Expanding the squared norms and rearranging, it follows that (cf. Wainwright (2019, Equation 7.29))

$$\frac{1}{n} \sum_{j=1}^{n} \left( \left\| \hat{\theta}_{\lambda_1,\lambda_2}^\top \Phi_j \right\|_{\mathcal{L}^2(S,\mathfrak{m})}^2 - \left\| \theta_*^\top \Phi_j \right\|_{\mathcal{L}^2(S,\mathfrak{m})}^2 \right) + \lambda_2 \left( \left\| \hat{\theta}_{\lambda_1,\lambda_2} \right\|^2 - \left\| \theta_* \right\|^2 \right)$$

$$\leq \frac{2}{n} \sum_{j=1}^{n} \left\langle \mathrm{I}_{y^{(j)}}, \Delta^\top \Phi_j \right\rangle + \lambda_1 \left( \left\| \theta_* \right\|_1 - \left\| \hat{\theta}_{\lambda_1,\lambda_2} \right\|_1 \right) . \tag{31}$$

Combining Equations (30) and (31), we obtain

$$\left\| \Delta \right\|_{U_n + \lambda_2 I_d}^2 \leq \underbrace{\frac{2}{n} \sum_{j=1}^{n} \left\langle \mathrm{I}_{y^{(j)}}, \Delta^\top \Phi_j \right\rangle - \frac{2}{n} \sum_{j=1}^{n} \left\langle \Delta^\top \Phi_j, \theta_*^\top \Phi_j \right\rangle - 2\lambda_2 \Delta^\top \theta_*}_{①} + \lambda_1 \underbrace{\left( \left\| \theta_* \right\|_1 - \left\| \hat{\theta}_{\lambda_1,\lambda_2} \right\|_1 \right)}_{②} . \tag{32}$$

Next, we upper-bound ①. We have

$$① \overset{(a)}{=} \frac{2}{n} \sum_{j=1}^{n} \left\langle \mathrm{I}_{y^{(j)}} - \theta_*^\top \Phi_j, \Delta^\top \Phi_j \right\rangle - 2\lambda_2 \Delta^\top \theta_*$$

$$\overset{(b)}{=} \Delta^\top \left( \frac{2}{n} \sum_{j=1}^{n} \left\langle \mathrm{I}_{y^{(j)}} - \theta_*^\top \Phi_j, \Phi_j \right\rangle \right) - 2\lambda_2 \Delta^\top \theta_* \tag{33}$$

$$\overset{(c)}{=} \Delta^\top \left( \frac{2}{n} \sum_{j=1}^{n} \left\langle \mathrm{I}_{y^{(j)}} - \theta_*^\top \Phi_j, \Phi_j \right\rangle \right) - 2\lambda_2 \left[ \Delta \right]_{S_*}^\top \left[ \theta_* \right]_{S_*}$$

$$\overset{(d)}{\leq} \left\| \Delta \right\|_1 \left\| \frac{2}{n} \sum_{j=1}^{n} \left\langle \mathrm{I}_{y^{(j)}} - \theta_*^\top \Phi_j, \Phi_j \right\rangle \right\|_\infty + 2\lambda_2 \left\| \left[ \Delta \right]_{S_*} \right\|_1 \left\| \theta_* \right\|_\infty$$

$$\overset{(e)}{\leq} \frac{\lambda_1}{2} \left\| \Delta \right\|_1 + 2\lambda_2 \left\| [\Delta]_{S_*} \right\|_1 \left\| \theta_* \right\|_\infty$$

$$\overset{(f)}{\leq} \frac{\lambda_1}{2} \left\| \Delta \right\|_1 + 2\lambda_2 \left\| [\Delta]_{S_*} \right\|_1 , \tag{34}$$

where (a) and (b) hold by the linearity of inner product, (c) holds because $\mathrm{supp}(\theta_*) = S_*$, (d) follows from Hölder's inequality, (e) holds by substituting in (29), and (f) holds because $\theta_*$ is a PMF. Next, we upper-bound ②. We have

$$② \overset{(a)}{=} \sum_{i=1}^{d} |[\theta_*]_i| - \sum_{i=1}^{d} |[\theta_* + \Delta]_i|$$

$$\overset{(b)}{=} \sum_{i \in S_*} |[\theta_*]_i| - \sum_{i \in S_*} |[\theta_* + \Delta]_i| - \sum_{i \in S_*^c} |[\Delta]_i|$$

$$\overset{(c)}{=} \left\| [\theta_*]_{S_*} \right\|_1 - \left\| [\theta_*]_{S_*} + [\Delta]_{S_*} \right\|_1 - \left\| [\Delta]_{S_*^c} \right\|_1$$

$$\overset{(d)}{\leq} \left\| [\Delta]_{S_*} \right\|_1 - \left\| [\Delta]_{S_*^c} \right\|_1 , \tag{35}$$

where (a) holds by definition of $\ell^1$-norm and $\Delta$, (b) holds because $\mathrm{supp}(\theta_*) = S_*$, (c) holds by definition of $\ell^1$-norm, and (d) holds by the triangle inequality. Combining Equations (32), (34) and (35), we obtain

$$\left\| \Delta \right\|_{U_n + \lambda_2 I_d}^2 \leq \left( \frac{3\lambda_1}{2} + 2\lambda_2 \right) \left\| [\Delta]_{S_*} \right\|_1 - \frac{\lambda_1}{2} \left\| [\Delta]_{S_*^c} \right\|_1 \tag{36}$$

$$\leq \left( \frac{3\lambda_1}{2} + 2\lambda_2 \right) \left\| [\Delta]_{S_*} \right\|_1$$

$$\overset{(a)}{\leq} \left( \frac{3\lambda_1}{2} + 2\lambda_2 \right) \sqrt{s} \left\| [\Delta]_{S_*} \right\|$$

$$\leq \left( \frac{3\lambda_1}{2} + 2\lambda_2 \right) \sqrt{s} \left\| \Delta \right\|$$

$$\overset{(b)}{\leq} \left( \frac{3\lambda_1}{2} + 2\lambda_2 \right) \sqrt{\frac{s}{\kappa + \lambda_2}} \left\| \Delta \right\|_{U_n + \lambda_2 I_d} ,$$

where (a) holds by the equivalence between $\ell^1$- and $\ell^2$-norms, and (b) follows from the restricted eigenvalue condition on $U_n$ since rearranging (36) yields $\| [\Delta]_{S_*^c} \|_1 \leq (3 + 4\lambda_2/\lambda_1) \| [\Delta]_{S_*} \|_1$. Thus, the estimation error satisfies the bounds

$$\left\| \Delta \right\|_{U_n + \lambda_2 I_d} \leq \left( \frac{3\lambda_1}{2} + 2\lambda_2 \right) \sqrt{\frac{s}{\kappa + \lambda_2}}$$

$$\overset{(a)}{=} \left( 6 \sqrt{\frac{2}{n} \log\left( \frac{2d}{\delta} \right)} + 2\lambda_2 \right) \sqrt{\frac{s}{\kappa + \lambda_2}} , \tag{37}$$

where (a) holds by substituting in $\lambda_1$, and

$$\left\| \Delta \right\| \overset{(b)}{\leq} \frac{1}{\sqrt{\kappa + \lambda_2}} \left\| \Delta \right\|_{U_n + \lambda_2 I_d} = \left( 6 \sqrt{\frac{2}{n} \log\left( \frac{2d}{\delta} \right)} + 2\lambda_2 \right) \frac{\sqrt{s}}{\kappa + \lambda_2} ,$$

where (b) follows from the restricted eigenvalue condition on $U_n$. Finally, choosing $\lambda_2 = 3\sqrt{(2/n) \log(2d/\delta)} - 2\kappa$ results in

$$\left\| \Delta \right\|_{U_n + \lambda_2 I_d} \overset{(a)}{\leq} \left( 12 \sqrt{\frac{2}{n} \log\left( \frac{2d}{\delta} \right)} - 4\kappa \right) \sqrt{s} \left( 3 \sqrt{\frac{2}{n} \log\left( \frac{2d}{\delta} \right)} - \kappa \right)^{-\frac{1}{2}}$$

$$= 4\sqrt{s}\left(3\sqrt{\frac{2}{n}\log\left(\frac{2d}{\delta}\right)} - \kappa\right)^{\frac{1}{2}}$$

$$\overset{(b)}{\leq} 4\sqrt[4]{\frac{18s^2}{n}\log\left(\frac{2d}{\delta}\right)}$$

as desired, where (a) holds by substituting $\lambda_2$ into (37) and is well-defined because $\kappa < 3\sqrt{(2/n)\log(2d/\delta)}$, and (b) holds because $\kappa \geq 0$. ∎

Next, we prove Theorem 4.

*Proof of Theorem 4.* Let $\mathcal{E}$ be the event that $U_n$ satisfies the $(\kappa/2, 3+4\lambda_2/\lambda_1)$-restricted eigenvalue condition over $S_*$. Invoking Lemma 2 with $\delta := \delta_1$ and $\gamma := 3 + 4\lambda_2/\lambda_1$, we get

$$\mathbb{P}(\mathcal{E}) \geq 1 - \delta_1. \tag{38}$$

Next, let $\mathcal{F}$ be the event that the estimation error satisfies the desired bounds

$$\left\|\hat{\theta}_{\lambda_1,\lambda_2} - \theta_*\right\|_{U_n+\lambda_2 I_d} \leq \left(6\sqrt{\frac{2}{n}\log\left(\frac{2d}{\delta_2}\right)} + 2\lambda_2\right)\sqrt{\frac{s}{(\kappa/2)+\lambda_2}}$$

$$= \left(6\sqrt{\frac{2}{n}\log\left(\frac{2d}{\delta_2}\right)} + 2\lambda_2\right)\sqrt{\frac{2s}{\kappa+2\lambda_2}}, \tag{39}$$

$$\left\|\hat{\theta}_{\lambda_1,\lambda_2} - \theta_*\right\| \leq \left(6\sqrt{\frac{2}{n}\log\left(\frac{2d}{\delta_2}\right)} + 2\lambda_2\right)\frac{\sqrt{s}}{(\kappa/2)+\lambda_2}$$

$$= \left(6\sqrt{\frac{2}{n}\log\left(\frac{2d}{\delta_2}\right)} + 2\lambda_2\right)\frac{2\sqrt{s}}{\kappa+2\lambda_2}.$$

Invoking Theorem 3 with $\kappa := \kappa/2$ and $\delta := \delta_2$, we get $\mathbb{P}(\mathcal{F}|\mathcal{E}) \geq 1 - \delta_2$. Thus,

$$\mathbb{P}(\mathcal{F}) \geq \mathbb{P}(\mathcal{E} \cap \mathcal{F}) = \mathbb{P}(\mathcal{E})\,\mathbb{P}(\mathcal{F} \mid \mathcal{E}) \geq (1-\delta_1)(1-\delta_2) \geq 1 - \delta_1 - \delta_2$$

as desired. Finally, choosing $\lambda_2 = 3\sqrt{(2/n)\log(2d/\delta_2)} - \kappa$ results in

$$\left\|\hat{\theta}_{\lambda_1,\lambda_2} - \theta_*\right\|_{U_n+\lambda_2 I_d} \overset{(a)}{\leq} \left(12\sqrt{\frac{2}{n}\log\left(\frac{2d}{\delta_2}\right)} - 2\kappa\right)\sqrt{2s}\left(6\sqrt{\frac{2}{n}\log\left(\frac{2d}{\delta_2}\right)} - \kappa\right)^{-\frac{1}{2}}$$

$$= 2\sqrt{2s}\left(6\sqrt{\frac{2}{n}\log\left(\frac{2d}{\delta_2}\right)} - \kappa\right)^{\frac{1}{2}}$$

$$\overset{(b)}{\leq} 4\sqrt[4]{\frac{18s^2}{n}\log\left(\frac{2d}{\delta_2}\right)}$$

as desired, where (a) holds by substituting $\lambda_2$ into (39) and is well-defined because $\kappa < 6\sqrt{(2/n)\log(2d/\delta_2)}$, and (b) holds because $\kappa \geq 0$. ∎

## D  Proofs of Dantzig Selector Upper Bounds

In this section, we prove Theorems 5 and 6.

*Proof of Theorem 5.* Throughout this proof, we restrict to the subset of the probability space where

$$\left\| \frac{1}{n} \sum_{j=1}^{n} \left\langle I_{y^{(j)}} - \theta_*^\top \Phi_j, \Phi_j \right\rangle \right\|_\infty \leq \sqrt{\frac{2}{n} \log\left(\frac{2d}{\delta}\right)} = \lambda, \tag{40}$$

which holds with probability at least $1 - \delta$ by Lemma 1. Our proof uses the following result which we distill from the statistics literature and restate for convenience:

**Lemma 4** (Candes & Tao (2007, Lemma 3.1))**.** *Let $A \in \mathbb{R}^{d \times d}$ be a symmetric matrix. Assume $A$ satisfies the $(\epsilon, p)$-restricted isometry property and the $(\zeta, p, 2p)$-restricted orthogonality property with $\epsilon + \zeta < 1$ and $p \leq d/2$. Fix $v \in \mathbb{R}^d$ and $S \subset [d]$ with $|S| = p$. Let $T = S \cup \arg_p \max_{i \in [d]-S} |[v]_i|$. (Refer to Appendix A for the definition of $\arg_p \max_{i \in [d]-S}$.) Then,*

$$\|[v]_T\| \leq \frac{1}{1 - \epsilon} \left\| [A]_{\langle T \rangle} v \right\| + \frac{\zeta}{(1 - \epsilon)\sqrt{p}} \|[v]_{S^c}\|_1, \tag{41}$$

$$\|v\|^2 \leq \|[v]_T\|^2 + \frac{1}{p} \|[v]_{S^c}\|_1^2. \tag{42}$$

For notational simplicity, let $\Delta = \bar{\theta}_\lambda - \theta_*$ and $S_\dagger = S_* \cup \arg_s \max_{i \in [d]-S_*} |[\Delta]_i|$. Invoking Lemma 4 with $A := U_n$, $v := \Delta$, and $S := S_*$, it follows from (42) that

$$\|\Delta\| \leq \sqrt{\left\| [\Delta]_{S_\dagger} \right\|^2 + \frac{1}{s} \left\| [\Delta]_{S_*^c} \right\|_1^2}. \tag{43}$$

Next, we upper-bound $\|[\Delta]_{S_*^c}\|_1$. Observe that

$$\begin{aligned}
\left\| [\Delta]_{S_*^c} \right\|_1 &= \sum_{i \notin S_*} |[\Delta]_i| \\
&\overset{(a)}{\leq} \sum_{i \notin S_*} |[\Delta]_i| + \sum_{i \in S_*} |[\theta_* + \Delta]_i| - \left( \sum_{i \in S_*} |[\theta_*]_i| - \sum_{i \in S_*} [\Delta]_i \right) \\
&\overset{(b)}{=} \sum_{i=1}^{d} |[\theta_* + \Delta]_i| - \left( \sum_{i=1}^{d} |[\theta_*]_i| - \sum_{i \in S_*} [\Delta]_i \right) \\
&= \|\theta_* + \Delta\|_1 + \left\| [\Delta]_{S_*} \right\|_1 - \|\theta_*\|_1 \\
&\overset{(c)}{=} \left\| \bar{\theta}_\lambda \right\|_1 + \left\| [\Delta]_{S_*} \right\|_1 - \|\theta_*\|_1 \\
&\overset{(d)}{\leq} \left\| [\Delta]_{S_*} \right\|_1 \\
&\overset{(e)}{\leq} \sqrt{s} \left\| [\Delta]_{S_*} \right\| \\
&\leq \sqrt{s} \left\| [\Delta]_{S_\dagger} \right\|, \tag{44}
\end{aligned}$$

where (a) holds by the reverse triangle inequality, (b) holds because $\mathrm{supp}(\theta_*) = S_*$, (c) holds by definition of $\Delta$, (d) holds because $\bar{\theta}_\lambda$ and $\theta_*$ both satisfy the constraint (6) and $\bar{\theta}_\lambda$ minimizes the $\ell^1$-norm among all such vectors by (5), and (e) holds by the equivalence between $\ell^1$- and $\ell^2$-norms. Combining Equations (43) and (44), we obtain

$$\|\Delta\| \leq \sqrt{2} \left\| [\Delta]_{S_\dagger} \right\|. \tag{45}$$

Next, we upper-bound $\|[\Delta]_{S_\dagger}\|$. We have

$$\left\| [\Delta]_{S_\dagger} \right\| \overset{(a)}{\leq} \frac{1}{1 - \epsilon} \left\| [U_n]_{\langle S_\dagger \rangle} \Delta \right\| + \frac{\zeta}{(1 - \epsilon)\sqrt{s}} \left\| [\Delta]_{S_*^c} \right\|_1$$

$$\overset{(b)}{\leq} \frac{1}{1-\epsilon} \left\| [U_n]_{\langle S_\dagger \rangle} \Delta \right\| + \frac{\zeta}{1-\epsilon} \left\| [\Delta]_{S_\dagger} \right\|,$$

where (a) holds by (41) and (b) holds by (44). Rearranging, it follows that

$$\left\| [\Delta]_{S_\dagger} \right\| \leq \frac{1}{1-\epsilon-\zeta} \left\| [U_n]_{\langle S_\dagger \rangle} \Delta \right\|. \tag{46}$$

Next, we upper-bound $\|[U_n]_{\langle S_\dagger \rangle} \Delta\|$. We have

$$\left\| [U_n]_{\langle S_\dagger \rangle} \Delta \right\| \overset{(a)}{\leq} \sqrt{2s} \left\| [U_n]_{\langle S_\dagger \rangle} \Delta \right\|_\infty$$
$$\leq \sqrt{2s} \left\| U_n \Delta \right\|_\infty$$
$$\overset{(b)}{=} \sqrt{2s} \left\| \frac{1}{n} \sum_{j=1}^n \left\langle \Delta^\top \Phi_j, \Phi_j \right\rangle \right\|_\infty$$
$$\overset{(c)}{=} \sqrt{2s} \left\| \frac{1}{n} \sum_{j=1}^n \left( \left\langle \mathrm{I}_{y^{(j)}} - \theta_*^\top \Phi_j, \Phi_j \right\rangle - \left\langle \mathrm{I}_{y^{(j)}} - \bar{\theta}_\lambda^\top \Phi_j, \Phi_j \right\rangle \right) \right\|_\infty \tag{47}$$
$$\overset{(d)}{\leq} \sqrt{2s} \left( \left\| \frac{1}{n} \sum_{j=1}^n \left\langle \mathrm{I}_{y^{(j)}} - \theta_*^\top \Phi_j, \Phi_j \right\rangle \right\|_\infty + \left\| \frac{1}{n} \sum_{j=1}^n \left\langle \mathrm{I}_{y^{(j)}} - \bar{\theta}_\lambda^\top \Phi_j, \Phi_j \right\rangle \right\|_\infty \right) \tag{48}$$
$$\overset{(e)}{\leq} \sqrt{2s} \left( \lambda + \left\| \frac{1}{n} \sum_{j=1}^n \left\langle \mathrm{I}_{y^{(j)}} - \bar{\theta}_\lambda^\top \Phi_j, \Phi_j \right\rangle \right\|_\infty \right)$$
$$\overset{(f)}{\leq} 2\lambda\sqrt{2s}, \tag{49}$$

where (a) holds by the equivalence between $\ell^2$- and $\ell^\infty$-norms, (b) holds by definition of $U_n$, (c) holds by the linearity of inner product, (d) holds by the triangle inequality, (e) holds by substituting in (40), and (f) holds because $\bar{\theta}_\lambda$ satisfies the constraint (6). Combining Equations (45), (46) and (49) and substituting in $\lambda$, we obtain

$$\|\Delta\| \leq \frac{4\lambda\sqrt{s}}{1-\epsilon-\zeta} = \frac{4}{1-\epsilon-\zeta} \sqrt{\frac{2s}{n} \log\left(\frac{2d}{\delta}\right)}$$

as desired. ∎

Next, we prove Theorem 6.

*Proof of Theorem 6.* Let $\mathcal{E}$ be the event that

$$\mu_{\max}(U_n - \Sigma_n) \leq \min\{\epsilon, \zeta\} \quad \text{and} \quad \mu_{\min}(U_n - \Sigma_n) \geq -\min\{\epsilon, \zeta\}. \tag{50}$$

Under this event, $U_n$ satisfies the $(2\epsilon, 2s)$-restricted isometry property, because for any $v \in \mathbb{R}^d$ with $\|v\|_0 \leq 2s$,

$$v^\top U_n v = v^\top \Sigma_n v + v^\top (U_n - \Sigma_n) v$$
$$\leq v^\top \Sigma_n v + \mu_{\max}(U_n - \Sigma_n) \|v\|^2$$
$$\overset{(a)}{\leq} (1+\epsilon) \|v\|^2 + \mu_{\max}(U_n - \Sigma_n) \|v\|^2$$
$$\overset{(b)}{\leq} (1+2\epsilon) \|v\|^2,$$

where (a) holds by the restricted isometry property on $\Sigma_n$ and (b) holds by (50), and

$$v^\top U_n v \geq v^\top \Sigma_n v + \mu_{\min}(U_n - \Sigma_n) \|v\|^2$$

$$\overset{(c)}{\geq} (1 - \epsilon) \|v\|^2 + \mu_{\min}(U_n - \Sigma_n) \|v\|^2$$

$$\overset{(d)}{\geq} (1 - 2\epsilon) \|v\|^2 ,$$

where (c) holds by the restricted isometry property on $\Sigma_n$ and (b) holds by (50). Furthermore, $U_n$ also satisfies the $(2\zeta, s, 2s)$-restricted orthogonality property under event $\mathcal{E}$, because for any $u, v \in \mathbb{R}^d$ such that $\|u\|_0 \leq p$, $\|v\|_0 \leq q$, and $\mathrm{supp}(u) \cap \mathrm{supp}(p) = \emptyset$,

$$
\begin{aligned}
\left| u^\top U_n v \right| &= \left| u^\top \Sigma_n v + u^\top (U_n - \Sigma_n) v \right| \\
&\overset{(a)}{\leq} \left| u^\top \Sigma_n v \right| + \left| u^\top (U_n - \Sigma_n) v \right| \\
&\overset{(b)}{\leq} \left| u^\top \Sigma_n v \right| + \|u\| \|U_n - \Sigma_n v\| \\
&\leq \left| u^\top \Sigma_n v \right| + \max \left\{ \mu_{\max}(U_n - \Sigma_n), -\mu_{\min}(U_n - \Sigma_n) \right\} \|u\| \|v\| \\
&\overset{(c)}{\leq} \zeta \|u\| \|v\| + \max \left\{ \mu_{\max}(U_n - \Sigma_n), -\mu_{\min}(U_n - \Sigma_n) \right\} \|u\| \|v\| \\
&\overset{(d)}{\leq} 2\zeta \|u\| \|v\| ,
\end{aligned}
$$

where (a) holds by the triangle inequality, (b) holds by the Cauchy-Schwarz inequality, (c) holds by the restricted orthogonality property on $\Sigma_n$, and (d) holds by (50). The probability of event $\mathcal{E}$ is

$$
\begin{aligned}
\mathbb{P}(\mathcal{E}) &\overset{(a)}{\geq} 1 - \mathbb{P}(\mu_{\max}(U_n - \Sigma_n) \geq \min \{\epsilon, \zeta\}) - \mathbb{P}(\mu_{\min}(U_n - \Sigma_n) \leq -\min \{\epsilon, \zeta\}) \\
&\overset{(b)}{\geq} 1 - 2d \exp \left( -\frac{n^2 \min \{\epsilon, \zeta\}^2}{8nd^2} \right) \\
&\overset{(c)}{\geq} 1 - \delta_1 ,
\end{aligned}
\tag{51}
$$

where (a) holds by the union bound, (b) holds by Equations (24) and (25) by the argument in (27) after invoking Lemma 3 with $\tau := \min\{\epsilon, \zeta\}$, and (c) holds because $n \geq (8d^2 / \min\{\epsilon, \zeta\}^2) \log(2d/\delta_1)$.

Next, let $\mathcal{F}$ be the event that the estimation error satisfies the desired bound

$$\left\| \bar{\theta}_\lambda - \theta_* \right\| \leq \frac{4}{1 - 2\epsilon - 2\zeta} \sqrt{\frac{2s}{n} \log \left( \frac{2d}{\delta_2} \right)} .$$

Invoking Theorem 5 with $\epsilon := 2\epsilon$, $\zeta := 2\zeta$, and $\delta := \delta_2$, we get $\mathbb{P}(\mathcal{F}|\mathcal{E}) \geq 1 - \delta_2$. Thus,

$$\mathbb{P}(\mathcal{F}) \geq \mathbb{P}(\mathcal{E} \cap \mathcal{F}) = \mathbb{P}(\mathcal{E}) \mathbb{P}(\mathcal{F} \mid \mathcal{E}) \geq (1 - \delta_1)(1 - \delta_2) \geq 1 - \delta_1 - \delta_2$$

as desired. ∎

## E   Proof of Lower Bound

In this section, we prove Corollary 1.

*Proof.* We have

$$
\mathfrak{R}\left( \theta \left( \mathcal{P}_{x^{1:n}}^{d,s} \right) \right) \geq \inf_{\hat{\theta} \in \hat{\Theta}_{d,n}} \sup_{P \in \mathcal{P}_{x^{1:n}}^{d,s}} \mathbb{E}_{Y^{1:n} \sim P} \left[ \left\| \left[ \hat{\theta}(Y^{1:n}) \right]_{S_*(P)} - [\theta(P)]_{S_*(P)} \right\| \right]
$$

$$
\overset{(a)}{=} \inf_{\hat{\theta} \in \hat{\Theta}_{d,n}} \sup_{\substack{S \subseteq [d] \\ |S| = s}} \sup_{\substack{P \in \mathcal{P}_{x^{1:n}}^{d,s} \\ S_*(P) = S}} \mathbb{E}_{Y^{1:n} \sim P} \left[ \left\| \left[ \hat{\theta}(Y^{1:n}) \right]_S - [\theta(P)]_{S_*(P)} \right\| \right]
$$

$$
\overset{(b)}{=} \inf_{\substack{\hat{\theta} \in \hat{\Theta}_{d,n} \\ S \subseteq [d] \\ |S| = s}} \sup_{Q \in \mathcal{P}^s_{x^{1:n}}} \sup_{Y^{1:n} \sim Q} \mathbb{E} \left[ \left\| \left[ \hat{\theta}\big(Y^{1:n}\big) \right]_S - \theta(Q) \right\| \right]
$$

$$
\geq \inf_{\substack{\hat{\theta} \in \hat{\Theta}_{d,n} \\ |S| = s}} \inf_{\substack{S \subseteq [d]}} \sup_{Q \in \mathcal{P}^s_{x^{1:n}}} \sup_{Y^{1:n} \sim Q} \mathbb{E} \left[ \left\| \left[ \hat{\theta}\big(Y^{1:n}\big) \right]_S - \theta(Q) \right\| \right]
$$

$$
\overset{(c)}{=} \inf_{\tilde{\theta} \in \hat{\Theta}_{s,n}} \sup_{Q \in \mathcal{P}^s_{x^{1:n}}} \mathbb{E}_{Y^{1:n} \sim Q} \left[ \left\| \tilde{\theta}\big(Y^{1:n}\big) - \theta(Q) \right\| \right], \tag{52}
$$

where in (a) we partition the distributions in $\mathcal{P}^{d,s}_{x^{1:n}}$ based on the support of their parameter, (b) holds because every distribution $P \in \mathcal{P}^{d,s}_{x^{1:n}}$ has an equivalent distribution $Q \in \mathcal{P}^s_{x^{1:n}}$ with $\theta(Q) = [\theta(P)]_{S_*(P)}$, and (c) holds because every estimator $\hat{\theta} \in \hat{\Theta}_{d,n}$ and set $S \subseteq [d]$ with cardinality $|S| = s$ defines an estimator $\tilde{\theta}(\cdot) = [\hat{\theta}(\cdot)]_S \in \hat{\Theta}_{s,n}$. Our proof uses the following result which we distill from the literature and restate for convenience:

**Lemma 5** (Zhang et al. (2024, Theorem 8)). *For any $d \in \mathbb{N}$, $n \in \mathbb{N}$, and $x^{1:n} \in \mathcal{X}^n$, the minimax $\ell^2$-risk of general contextual CDF regression satisfies the lower bound*

$$
\mathfrak{R}\big(\theta\big(\mathcal{P}^d_{x^{1:n}}\big)\big) = \inf_{\hat{\theta} \in \hat{\Theta}_{d,n}} \sup_{P \in \mathcal{P}^d_{x^{1:n}}} \mathbb{E}_{Y^{1:n} \sim P} \left[ \left\| \hat{\theta}\big(Y^{1:n}\big) - \theta(P) \right\| \right]
$$

$$
\geq \Omega \left( \min \left\{ 1, \sqrt{\frac{d}{1 + n \sup_{P \in \mathcal{P}^d_{x^{1:n}}} \mu_{\min}(U_n(P))}} \right\} \right).
$$

Invoking Lemma 5 with $d := s$ and combining the result with (52), we obtain

$$
\mathfrak{R}\Big(\theta\Big(\mathcal{P}^{d,s}_{x^{1:n}}\Big)\Big) \geq \Omega \left( \min \left\{ 1, \sqrt{\frac{s}{1 + n \sup_{Q \in \mathcal{P}^s_{x^{1:n}}} \mu_{\min}(U_n(Q))}} \right\} \right). \tag{53}
$$

Next, we upper-bound $\mu_{\min}(U_n(Q))$ for any $Q \in \mathcal{P}^s_{x^{1:n}}$. Observe that

$$
\mu_{\min}(U_n(Q)) \overset{(a)}{\leq} \frac{1}{s} \operatorname{Tr}(U_n(Q))
$$

$$
\overset{(b)}{=} \frac{1}{s} \sum_{i=1}^{s} [U_n(Q)]_{i,i}
$$

$$
\overset{(c)}{=} \frac{1}{s} \sum_{i=1}^{s} \frac{1}{n} \sum_{j=1}^{n} \int_S \phi_i\big(x^{(j)}, t\big)^2 d\mathfrak{m}
$$

$$
\overset{(d)}{\leq} \frac{1}{s} \sum_{i=1}^{s} \frac{1}{n} \sum_{j=1}^{n} \mathfrak{m}(S) \tag{54}
$$

$$
\overset{(e)}{=} 1, \tag{55}
$$

where (a) holds because the trace of a matrix is the sum of its eigenvalues, (b) holds because the trace of a matrix is the sum of its diagonal entries, (c) holds by definition of $U_n$, (d) holds because $\phi_i(x^{(j)}, \cdot)$ is a CDF, and (e) holds because $\mathfrak{m}$ is a probability measure. Combining Equations (53) and (55), we obtain

$$
\mathfrak{R}\Big(\theta\Big(\mathcal{P}^{d,s}_{x^{1:n}}\Big)\Big) \geq \Omega \left( \min \left\{ 1, \sqrt{\frac{s}{1 + n}} \right\} \right)
$$

$$
\geq \Omega \left( \min \left\{ 1, \sqrt{\frac{s}{2n}} \right\} \right)
$$

$$\stackrel{(a)}{=} \Omega\left(\sqrt{\frac{s}{n}}\right)$$

as desired, where (a) holds because $n \geq s/2$. ■

## F    Examples

In this section, we present three examples of contextual CDF bases which non-trivially satisfy the restricted eigenvalue condition (i.e., with $\kappa > 0$ in Definition 1).

### F.1    Categorical CDFs

Our first example considers the expected Gramian matrix $\Sigma_n$ of a family of categorical CDFs drawn independently from a fixed set of Dirichlet priors:

**Proposition 1** (Categorical CDFs). *Fix $d \in \mathbb{N}$ Dirichlet distributions $\mathsf{Dirichlet}(\alpha_{i,1}, \ldots, \alpha_{i,K})$ indexed by $i \in [d]$. For each Dirichlet distribution, let $\alpha_{i,0} = \sum_{k=1}^{K} \alpha_{i,k}$ denote the sum of its parameters and let $\tilde{\alpha}_{i,k} = \alpha_{i,k}/\alpha_{i,0}$ denote its normalized $k$th parameter. Assume $n \in \mathbb{N}$ i.i.d. context samples are generated, where each context sample $x^{(j)} = (x_1^{(j)}, \ldots, x_d^{(j)})$ consists of $d$ categorical PMFs sampled independently from the respective Dirichlet priors, i.e.,*

$$\forall i \in [d], \ x_i^{(j)} = \left(x_{i,1}^{(j)}, \ldots, x_{i,K}^{(j)}\right) \sim \mathsf{Dirichlet}(\alpha_{i,1}, \ldots, \alpha_{i,K}).$$

*Given a context sample $x$ as defined above, let the $i$th contextual basis function $\phi_i(x, t)$ be the CDF of a $\mathsf{Categorical}(x_{i,1}, \ldots, x_{i,K})$ random variable. Let $\mathfrak{m}$ be the uniform measure over the support $S = [1, K]$. Then the matrix $\Sigma_n$ satisfies the $(\kappa, \gamma)$-restricted eigenvalue condition over $S_*$ for*

$$\kappa = \frac{1}{K-1} \min_{i \in [d]} \left\{ \frac{1}{\alpha_{i,0}+1} \sum_{k=1}^{K-1} \left(\sum_{\ell=1}^{k} \tilde{\alpha}_{i,\ell}\right) \left(\sum_{\ell=k+1}^{K} \tilde{\alpha}_{i,\ell}\right) \right\}$$

*and arbitrary $\gamma \geq 0$ and $S_* \subseteq [d]$.*

*Proof.* The entries of $\Sigma_n$ are

$$
\begin{aligned}
[\Sigma_n]_{i,i'} &\stackrel{(a)}{=} \mathbb{E}\left[ \frac{1}{n} \sum_{j=1}^{n} \int_S \phi_i\left(X^{(j)}, t\right) \phi_{i'}\left(X^{(j)}, t\right) d\mathfrak{m} \right] \\
&\stackrel{(b)}{=} \mathbb{E}\left[ \int_S \phi_i(X, t)\, \phi_{i'}(X, t)\, d\mathfrak{m} \right] \\
&\stackrel{(c)}{=} \mathbb{E}\left[ \frac{1}{K-1} \int_1^K \phi_i(X, t)\, \phi_{i'}(X, t)\, dt \right] \\
&\stackrel{(d)}{=} \mathbb{E}\left[ \frac{1}{K-1} \sum_{k=1}^{K-1} \int_k^{k+1} \left(\sum_{\ell=1}^{k} X_{i,\ell}\right) \left(\sum_{\ell=1}^{k} X_{i',\ell}\right) dt \right] \\
&= \frac{1}{K-1} \sum_{k=1}^{K-1} \mathbb{E}\left[ \left(\sum_{\ell=1}^{k} X_{i,\ell}\right) \left(\sum_{\ell=1}^{k} X_{i',\ell}\right) \right],
\end{aligned}
$$

where (a) holds by definition of $\Sigma_n$, (b) holds because the context samples $X^{(j)}$ are i.i.d., (c) follows from the support $S = [1, K]$ and the choice of measure, and (d) holds by splitting the domain of integration and substituting in the definition of $\phi_i$. The diagonal entries of $\Sigma_n$ (i.e., $i = i'$) are

$$[\Sigma_n]_{i,i} = \frac{1}{K-1} \sum_{k=1}^{K-1} \left( \sum_{\ell=1}^{k} \mathbb{E}\left[X_{i,\ell}^2\right] + \sum_{\ell=1}^{k} \sum_{\substack{\ell'=1 \\ \ell' \neq \ell}}^{k} \mathbb{E}[X_{i,\ell} X_{i,\ell'}] \right)$$

$$\overset{(a)}{=} \frac{1}{K-1} \sum_{k=1}^{K-1} \left( \sum_{\ell=1}^{k} \frac{\alpha_{i,0}\tilde{\alpha}_{i,\ell}^2 + \tilde{\alpha}_{i,\ell}}{\alpha_{i,0}+1} + \sum_{\ell=1}^{k} \sum_{\substack{\ell'=1 \\ \ell' \neq \ell}}^{k} \frac{\alpha_{i,0}}{\alpha_{i,0}+1} \tilde{\alpha}_{i,\ell}\tilde{\alpha}_{i,\ell'} \right)$$

$$= \frac{1}{K-1} \sum_{k=1}^{K-1} \left( \frac{\alpha_{i,0}}{\alpha_{i,0}+1} \left( \sum_{\ell=1}^{k} \tilde{\alpha}_{i,\ell} \right)^2 + \frac{1}{\alpha_{i,0}+1} \sum_{\ell=1}^{k} \tilde{\alpha}_{i,\ell} \right), \tag{56}$$

where (a) follows from the properties of the Dirichlet distribution. The off-diagonal entries of $\Sigma_n$ (i.e., $i \neq i'$) are

$$[\Sigma_n]_{i,i'} \overset{(a)}{=} \frac{1}{K-1} \sum_{k=1}^{K-1} \mathbb{E}\left[ \sum_{\ell=1}^{k} X_{i,\ell} \right] \mathbb{E}\left[ \sum_{\ell=1}^{k} X_{i',\ell} \right]$$

$$\overset{(b)}{=} \frac{1}{K-1} \sum_{k=1}^{K-1} \left( \sum_{\ell=1}^{k} \tilde{\alpha}_{i,\ell} \right) \left( \sum_{\ell=1}^{k} \tilde{\alpha}_{i',\ell} \right), \tag{57}$$

where (a) holds because the categorical distributions are pairwise independent and (b) follows from the properties of the Dirichlet distribution. Thus, for any $v \in \mathbb{R}^d$,

$$v^\top \Sigma_n v = \sum_{i=1}^{d} \sum_{i'=1}^{d} [\Sigma_n]_{i,i'} [v]_i [v]_{i'}$$

$$\overset{(a)}{=} \sum_{i=1}^{d} \frac{1}{K-1} \sum_{k=1}^{K-1} \left( \frac{\alpha_{i,0}}{\alpha_{i,0}+1} \left( \sum_{\ell=1}^{k} \tilde{\alpha}_{i,\ell} \right)^2 + \frac{1}{\alpha_{i,0}+1} \sum_{\ell=1}^{k} \tilde{\alpha}_{i,\ell} \right) [v]_i^2$$

$$+ \sum_{i=1}^{d} \sum_{\substack{i'=1 \\ i' \neq i}}^{d} \frac{1}{K-1} \sum_{k=1}^{K-1} \left( \sum_{\ell=1}^{k} \tilde{\alpha}_{i,\ell} \right) \left( \sum_{\ell=1}^{k} \tilde{\alpha}_{i',\ell} \right) [v]_i [v]_{i'}$$

$$= \frac{1}{K-1} \sum_{k=1}^{K-1} \left( \sum_{i=1}^{d} \frac{[v]_i^2}{\alpha_{i,0}+1} \left( \sum_{\ell=1}^{k} \tilde{\alpha}_{i,\ell} \right) \left( 1 - \sum_{\ell=1}^{k} \tilde{\alpha}_{i,\ell} \right) + \left( \sum_{i=1}^{d} [v]_i \sum_{\ell=1}^{k} \tilde{\alpha}_{i,\ell} \right)^2 \right)$$

$$\geq \frac{1}{K-1} \sum_{k=1}^{K-1} \sum_{i=1}^{d} \frac{[v]_i^2}{\alpha_{i,0}+1} \left( \sum_{\ell=1}^{k} \tilde{\alpha}_{i,\ell} \right) \left( \sum_{\ell=k+1}^{K} \tilde{\alpha}_{i,\ell} \right)$$

$$\geq \frac{1}{K-1} \min_{i \in [d]} \left\{ \frac{1}{\alpha_{i,0}+1} \sum_{k=1}^{K-1} \left( \sum_{\ell=1}^{k} \tilde{\alpha}_{i,\ell} \right) \left( \sum_{\ell=k+1}^{K} \tilde{\alpha}_{i,\ell} \right) \right\} \|v\|^2$$

$$= \kappa \|v\|^2$$

as desired, where (a) holds by substituting in Equations (56) and (57). $\blacksquare$

### F.2 Step CDFs

Our second example considers the empirical Gramian matrix $U_n$ of a family of shifted step-function CDFs:

**Proposition 2** (Step CDFs). *Fix $d \in \mathbb{N}$. Let $\mathcal{X}$ be the space of all d-tuples $x = (x_1, \ldots, x_d)$ such that $0 \leq x_1 < \cdots < x_d < 1$. Consider the contextual CDF basis*

$$\forall i \in [d], \quad \phi_i(x,t) = \mathbb{1}\{t \geq x_i\}.$$

*Let $\mathfrak{m}$ be the uniform measure over the support $S = [0,1]$. Then for any $n \in \mathbb{N}$ and $x^{1:n} \in \mathcal{X}^n$, the matrix $U_n$ satisfies the $(\kappa, \gamma)$-restricted eigenvalue condition over $S_*$ for*

$$\kappa = \frac{1}{4n} \sum_{j=1}^{n} \min_{i \in [d]} \left\{ x_{i+1}^{(j)} - x_i^{(j)} \right\}$$

*and arbitrary $\gamma \geq 0$ and $S_* \subseteq [d]$, where for notational simplicity we define $x_{d+1}^{(j)} = 1$ for each $j \in [n]$.*

*Proof.* For any $v \in \mathbb{R}^d$, we have

$$
\begin{aligned}
v^\top U_n v &\overset{(a)}{=} v^\top \left( \frac{1}{n} \sum_{j=1}^n \int_S \Phi_j \Phi_j^\top d\mathfrak{m} \right) v \\
&\overset{(b)}{=} \frac{1}{n} \sum_{j=1}^n \int_S \left( v^\top \Phi_j \right)^2 d\mathfrak{m} \\
&\overset{(c)}{=} \frac{1}{n} \sum_{j=1}^n \int_0^1 \left( v^\top \Phi_j \right)^2 dt \\
&\overset{(d)}{=} \frac{1}{n} \sum_{j=1}^n \int_0^1 \left( \sum_{i=1}^d [v]_i \, \mathbb{1}\left\{ t \geq x_i^{(j)} \right\} \right)^2 dt \\
&\overset{(e)}{=} \frac{1}{n} \sum_{j=1}^n \sum_{k=1}^d \int_{x_k^{(j)}}^{x_{k+1}^{(j)}} \left( \sum_{i=1}^k [v]_i \right)^2 dt \\
&= \frac{1}{n} \sum_{j=1}^n \sum_{k=1}^d \left( x_{k+1}^{(j)} - x_k^{(j)} \right) \left( \sum_{i=1}^k [v]_i \right)^2 \\
&= \frac{1}{n} \sum_{j=1}^n \sum_{k=1}^d \left( \sum_{i=1}^d \sqrt{x_{k+1}^{(j)} - x_k^{(j)}} \, \mathbb{1}\{i \leq k\} \, [v]_i \right)^2 ,
\end{aligned}
\tag{58}
$$

where (a) holds by definition of $U_n$, (b) holds by the linearity of integration, (c) follows from the support $S = [0,1]$ and the choice of measure, (d) holds by substituting in the definition of $\phi_i$, and (e) holds by splitting the domain of integration. For each $j \in [n]$, define a matrix $A_j \in \mathbb{R}^{d \times d}$ given by

$$
[A_j]_{k,i} = \sqrt{x_{k+1}^{(j)} - x_k^{(j)}} \, \mathbb{1}\{i \leq k\} .
\tag{59}
$$

Combining Equations (58) and (59), we obtain

$$
\begin{aligned}
v^\top U_n v &= \frac{1}{n} \sum_{j=1}^n \sum_{k=1}^d [A_j v]_k^2 \\
&= \frac{1}{n} \sum_{j=1}^n \| A_j v \|^2 \\
&\geq \frac{1}{n} \sum_{j=1}^n \sigma_{\min}(A_j)^2 \| v \|^2 \\
&= \frac{1}{n} \sum_{j=1}^n \frac{\| v \|^2}{\| A_j^{-1} \|^2} .
\end{aligned}
\tag{60}
$$

Next, we upper-bound $\| A_j^{-1} \|$. Since $A_j$ is lower triangular with identical values from the main diagonal leftwards in each row, its inverse is the bidiagonal matrix $A_j^{-1} = D_j + C_j$ given by

$$
[D_j]_{i,i'} = \left( x_{i+1}^{(j)} - x_i^{(j)} \right)^{-\frac{1}{2}} \mathbb{1}\{i = i'\} ,
$$

$$
[C_j]_{i,i'} = - \left( x_{i+1}^{(j)} - x_i^{(j)} \right)^{-\frac{1}{2}} \mathbb{1}\{i = i' + 1\} .
$$

It follows that

$$
\begin{aligned}
\left\| A_j^{-1} \right\| &\overset{(a)}{\leq} \| D_j \| + \| C_j \| \\
&\overset{(b)}{=} \max_{i \in [d]} \left( x_{i+1}^{(j)} - x_i^{(j)} \right)^{-\frac{1}{2}} + \max_{i \in [d-1]} \left( x_{i+1}^{(j)} - x_i^{(j)} \right)^{-\frac{1}{2}} \\
&\leq 2 \max_{i \in [d]} \left( x_{i+1}^{(j)} - x_i^{(j)} \right)^{-\frac{1}{2}} \\
&= 2 \left( \min_{i \in [d]} \left\{ x_{i+1}^{(j)} - x_i^{(j)} \right\} \right)^{-\frac{1}{2}},
\end{aligned}
\tag{61}
$$

where (a) holds by the triangle inequality and (b) holds by the diagonal structure of $D_j$ and $C_j$. Combining Equations (60) and (61), we obtain

$$
\begin{aligned}
v^\top U_n v &\geq \frac{1}{4n} \sum_{j=1}^{n} \min_{i \in [d]} \left\{ x_{i+1}^{(j)} - x_i^{(j)} \right\} \| v \|^2 \\
&= \kappa \| v \|^2 .
\end{aligned}
$$

$\blacksquare$

### F.3 Polynomial CDFs

Our third example considers the empirical Gramian matrix $U_n$ of a family of shifted and scaled polynomial CDFs of varying degrees:

**Proposition 3** (Polynomial CDFs). *Fix $d \in \mathbb{N}$ and $a \in (0, 1/d]$. Let $\mathcal{X}$ be the space of all $d$-tuples $x = (x_1, \ldots, x_d)$ of pairs $x_i = (b_i, r_i) \in [0, 1] \times (0, \infty)$ such that $b_{i+1} - b_i \geq a$ for all $i \in [d-1]$ and $1 - b_d \geq a$. Consider the contextual CDF basis*

$$
\forall i \in [d], \quad \phi_i(x, t) = \begin{cases} 0, & \text{if } t < b_i, \\ \left( \frac{t - b_i}{a} \right)^{r_i}, & \text{if } t \in [b_i, b_i + a), \\ 1, & \text{if } t \geq b_i + a. \end{cases}
$$

*Let $\mathfrak{m}$ be the uniform measure over the support $S = [0, 1]$. Then for any $n \in \mathbb{N}$ and $x^{1:n} \in \mathcal{X}^n$, the matrix $U_n$ satisfies the $(\kappa, \gamma)$-restricted eigenvalue over $S_*$ for*

$$
\kappa = \frac{1}{n} \sum_{j=1}^{n} \left( \frac{1}{4} \min_{k \in [d]} \alpha_{j,k} - \max_{i \in [d]} \beta_{i,j} \right)
$$

*and arbitrary $\gamma \geq 0$ and $S_* \subseteq [d]$, where for notational simplicity we define*

$$
\alpha_{j,k} = \begin{cases} \frac{a \left( r_{k+1}^{(j)} - r_k^{(j)} \right)}{\left( r_k^{(j)} + 1 \right) \left( r_{k+1}^{(j)} + 1 \right)} + b_{k+1}^{(j)} - b_k^{(j)}, & \text{if } k < d, \\ \frac{a}{r_d^{(j)} + 1} + 1 - a - b_d^{(j)}, & \text{if } k = d, \end{cases}
$$

$$
\beta_{i,j} = \frac{a r_i^{(j)}}{\left( r_i^{(j)} + 1 \right) \left( 2 r_i^{(j)} + 1 \right)},
$$

*for all $i \in [d]$, $j \in [n]$, and $k \in [d]$.*

*Proof.* The entries of $U_n$ are

$$
[U_n]_{i,i'} \overset{(a)}{=} \frac{1}{n} \sum_{j=1}^{n} \int_S \phi_i \left( x^{(j)}, t \right) \phi_{i'} \left( x^{(j)}, t \right) d\mathfrak{m}
$$

$$\stackrel{(b)}{=} \frac{1}{n} \sum_{j=1}^{n} \int_0^1 \phi_i\left(x^{(j)}, t\right) \phi_{i'}\left(x^{(j)}, t\right) dt,$$

where (a) holds by definition of $U_n$ and (b) follows from the support $S = [0, 1]$ and the choice of measure. The diagonal entries of $U_n$ (i.e., $i = i'$) are

$$[U_n]_{i,i} = \frac{1}{n} \sum_{j=1}^{n} \int_0^1 \phi_i\left(x^{(j)}, t\right)^2 dt$$

$$\stackrel{(a)}{=} \frac{1}{n} \sum_{j=1}^{n} \left( \int_{b_i^{(j)}}^{b_i^{(j)}+a} \left(\frac{t - b_i^{(j)}}{a}\right)^{2r_i^{(j)}} dt + \int_{b_i^{(j)}+a}^{1} 1 \, dt \right)$$

$$= \frac{1}{n} \sum_{j=1}^{n} \left( \frac{a}{2r_i^{(j)} + 1} + 1 - b_i^{(j)} - a \right)$$

$$= \frac{1}{n} \sum_{j=1}^{n} \underbrace{\left( \frac{a}{r_i^{(j)} + 1} + 1 - b_i^{(j)} - a \right)}_{①} - \frac{1}{n} \sum_{j=1}^{n} \underbrace{\frac{a r_i^{(j)}}{\left(r_i^{(j)} + 1\right)\left(2r_i^{(j)} + 1\right)}}_{\beta_{i,j}},$$

where (a) holds by substituting in the definition of $\phi_i$. The off-diagonal entries of $U_n$ (i.e., $i < i'$) are

$$[U_n]_{i,i'} \stackrel{(a)}{=} \frac{1}{n} \sum_{j=1}^{n} \left( \int_{b_{i'}^{(j)}}^{b_{i'}^{(j)}+a} \left(\frac{t - b_{i'}^{(j)}}{a}\right)^{r_{i'}^{(j)}} dt + \int_{b_{i'}^{(j)}+a}^{1} 1 \, dt \right)$$

$$= \frac{1}{n} \sum_{j=1}^{n} \underbrace{\left( \frac{a}{r_{i'}^{(j)} + 1} + 1 - b_{i'}^{(j)} - a \right)}_{①},$$

where (a) holds by substituting in the definitions of $\phi_i$ and $\phi_{i'}$. Next, we characterize ①. For any $j \in [n]$ and $i, i' \in [d]$ such that $i \leq i'$, we have

$$① \stackrel{(a)}{=} \sum_{k=i'}^{d-1} \left( \frac{a}{r_k^{(j)} + 1} - \frac{a}{r_{k+1}^{(j)} + 1} + b_{k+1}^{(j)} - b_k^{(j)} \right) + \left( \frac{a}{r_d^{(j)} + 1} + 1 - a - b_d^{(j)} \right)$$

$$= \sum_{k=i'}^{d-1} \left( \frac{a\left(r_{k+1}^{(j)} - r_k^{(j)}\right)}{\left(r_k^{(j)} + 1\right)\left(r_{k+1}^{(j)} + 1\right)} + b_{k+1}^{(j)} - b_k^{(j)} \right) + \left( \frac{a}{r_d^{(j)} + 1} + 1 - a - b_d^{(j)} \right)$$

$$\stackrel{(b)}{=} \sum_{k=1}^{d-1} \underbrace{\left( \frac{a\left(r_{k+1}^{(j)} - r_k^{(j)}\right)}{\left(r_k^{(j)} + 1\right)\left(r_{k+1}^{(j)} + 1\right)} + b_{k+1}^{(j)} - b_k^{(j)} \right)}_{\alpha_{j,k}} \mathbb{1}\{k \geq i\}\, \mathbb{1}\{k \geq i'\} + \underbrace{\left( \frac{a}{r_d^{(j)} + 1} + 1 - a - b_d^{(j)} \right)}_{\alpha_{j,d}},$$

where (a) holds because the sum over $k$ telescopes and (b) holds because $i \leq i'$. For each $j \in [n]$, define matrices $A_j, B_j \in \mathbb{R}^{d \times d}$ given by

$$[A_j]_{k,i} = \sqrt{\alpha_{j,k}}\, \mathbb{1}\{k \geq i\},$$
$$[B_j]_{i,i'} = \beta_{i,j}\, \mathbb{1}\{i = i'\}.$$

Combining the above, we obtain $U_n = (1/n) \sum_{j=1}^{n} (A_j^\top A_j - B_j)$. Thus, for any $v \in \mathbb{R}^d$,

$$v^\top U_n v = \frac{1}{n} \sum_{j=1}^{n} \left( v^\top A_j^\top A_j v - v^\top B_j v \right)$$

$$\geq \frac{1}{n} \sum_{j=1}^{n} \left( \|A_j v\|^2 - \|v\| \|B_j v\| \right)$$

$$\geq \frac{1}{n} \sum_{j=1}^{n} \left( \sigma_{\min}(A_j)^2 \|v\|^2 - \|B_j\| \|v\|^2 \right)$$

$$= \frac{1}{n} \sum_{j=1}^{n} \left( \frac{1}{\left\| A_j^{-1} \right\|^2} - \|B_j\| \right) \|v\|^2 . \tag{62}$$

Next, we upper-bound $\|A_j^{-1}\|$. Since $A_j$ is lower triangular with identical values from the main diagonal leftwards in each row, its inverse is the bidiagonal matrix $A_j^{-1} = D_j + C_j$ given by

$$[D_j]_{i,k} = \frac{1}{\sqrt{\alpha_{j,k}}} \mathbb{1}\{i = k\} ,$$

$$[C_j]_{i,k} = -\frac{1}{\sqrt{\alpha_{j,k}}} \mathbb{1}\{i = k + 1\} .$$

It follows that

$$\left\| A_j^{-1} \right\| \overset{(a)}{\leq} \|D_j\| + \|C_j\|$$

$$\overset{(b)}{=} \max_{k \in [d]} \frac{1}{\sqrt{\alpha_{j,k}}} + \max_{k \in [d-1]} \frac{1}{\sqrt{\alpha_{j,k}}}$$

$$\leq 2 \max_{k \in [d]} \frac{1}{\sqrt{\alpha_{j,k}}}$$

$$= 2 \left( \min_{k \in [d]} \alpha_{j,k} \right)^{-\frac{1}{2}} , \tag{63}$$

where (a) holds by the triangle inequality and (b) holds by the diagonal structure of $D_j$ and $C_j$. Furthermore, $\|B_j\| = \max_{i \in [d]} \beta_{i,j}$ by the diagonal structure of $B_j$. Combining Equations (62) and (63), we obtain

$$v^\top U_n v \geq \frac{1}{n} \sum_{j=1}^{n} \left( \frac{1}{4} \min_{k \in [d]} \alpha_{j,k} - \max_{i \in [d]} \beta_{i,j} \right) \|v\|^2$$

$$= \kappa \|v\|^2 .$$

$$\blacksquare$$

Lastly, we present a specific instance of the polynomial CDF basis where $\kappa > 0$ is readily apparent. Let $d, n \in \mathbb{N}$ be arbitrary and let $a = 1/(d+1)$. For each $i \in [d]$ and $j \in [n]$, let $b_i^{(j)} = (i-1)/(d+1)$ and $r_i^{(j)} = i$. Then,

$$\frac{1}{4} \min_{k \in [d]} \alpha_{j,k} = \frac{1}{4} \min_{k \in [d]} \begin{cases} \frac{1}{(d+1)(k+1)(k+2)} + \frac{1}{d+1} , & \text{if } k < d \\ \frac{1}{(d+1)^2} + \frac{1}{d+1} , & \text{if } k = d \end{cases}$$

$$> \frac{1}{4(d+1)}$$

$$> \max_{i \in [d]} \frac{i}{(d+1)(i+1)(2i+1)}$$

$$= \max_{i \in [d]} \beta_{i,j}$$

for each $j \in [n]$, and so $\kappa > 0$ as desired.

# G   Computational Complexity

In this appendix, we discuss the computational complexity of computing lasso, elastic net, and Dantzig selector estimators. First, observe that after pre-computing the matrix $A := \sum_{j=1}^{n} \int_S \Phi_j \Phi_j^\top d\mathfrak{m} \in \mathbb{R}^{d \times d}$ and the vector $b := \sum_{j=1}^{n} \int_S \mathrm{I}_{y^{(j)}} \Phi_j d\mathfrak{m} \in \mathbb{R}^d$, our lasso estimator, elastic net estimator, and Dantzig selector can be directly computed using any method for the standard lasso estimator, elastic net estimator, and Dantzig selector in $\mathbb{R}^d$. Specifically, the standard lasso estimator, elastic net estimator, and Dantzig selector in $\mathbb{R}^d$ with feature matrix $X \in \mathbb{R}^{n \times d}$ and response vector $y \in \mathbb{R}^n$ solve

$$
\begin{aligned}
\tilde{\theta}_{\mathsf{Lasso}} &:= \arg \min_{\theta \in \mathbb{R}^d} \left\{ \frac{1}{n} \|y - X\theta\|^2 + \lambda \|\theta\|_1 \right\} \\
&= \arg \min_{\theta \in \mathbb{R}^d} \left\{ \frac{1}{n} \left( \theta^\top (X^\top X)\theta - 2\theta^\top (X^\top y) \right) + \lambda \|\theta\|_1 \right\}, \\
\tilde{\theta}_{\mathsf{Elastic}} &:= \arg \min_{\theta \in \mathbb{R}^d} \left\{ \frac{1}{n} \|y - X\theta\|^2 + \lambda_1 \|\theta\|_1 + \lambda_2 \|\theta\|^2 \right\} \\
&= \arg \min_{\theta \in \mathbb{R}^d} \left\{ \frac{1}{n} \left( \theta^\top (X^\top X + \lambda_2 I)\theta - 2\theta^\top (X^\top y) \right) + \lambda \|\theta\|_1 \right\},
\end{aligned}
$$

and

$$
\begin{aligned}
\tilde{\theta}_{\mathsf{Dantzig}} := \operatorname*{minimize}_{\theta \in \mathbb{R}^d} \quad & \|\theta\|_1 \\
\text{subject to} \quad & \frac{1}{n} \left\| X^\top y - X^\top X\theta \right\|_\infty \leq \lambda,
\end{aligned}
$$

respectively. Any algorithm $\mathcal{A}$ that solves the above problems only accesses the matrix $X^\top X \in \mathbb{R}^{d \times d}$ and the vector $X^\top y \in \mathbb{R}^d$ from the problems. Therefore, noticing the similarity in the structures of our problems and the above standard problems, by replacing $X^\top X$ with $A$ and $X^\top y$ with $b$ defined previously, we can solve our estimator directly using $\mathcal{A}$.

Now, we analyze the computational complexity for calculating $A$ and $b$, which consists of the evaluation of $O(nd^2)$ one-dimensional integrals. In practice, users can pick an appropriate numerical integration method based on the specific form of the contextual CDFs $\Phi$ if it is hard or impossible to analytically calculate the integrals. Here, we consider the standard trapezoidal rule for numerical integration. For $m$ grid points, the computational complexity of integrating using the trapezoidal rule is $O(m)$ (assuming single CDF evaluation is constant time). Thus, the total computational complexity for the pre-processing step is $O(mnd^2)$.

Next, we use the above reasoning to analyze the computational complexity of our estimators using existing algorithms for standard lasso, elastic net, and Dantzig selector problems. Lasso and elastic net estimators can be computed using the fast iterative shrinkage-thresholding algorithm (FISTA) (Beck & Teboulle, 2009) which iteratively optimizes the objective function based on gradient evaluations. In the pre-processing step of FISTA, we evaluate the matrix $A$ and the vector $b$, which takes $O(mnd^2)$ time as analyzed above. After the pre-processing step, each iteration of FISTA takes $O(d^2)$ computation time. Thus, for $k \in \mathbb{N}$ iterations of FISTA applied for lasso and elastic net estimators, the computational complexity is $O((mn + k)d^2)$, where $m$ is the number of grid points used for numerical integration. Furthermore, the convergence rate of FISTA is $O(1/k^2)$ (Zhao & Huo, 2023).

For the Dantzig selector, we also first calculate $A$ and $b$ in $O(mnd^2)$. Then, after introducing slack variables $u, v \in \mathbb{R}^d$, (5) can be recast as the following standard linear program (Li et al., 2013):

$$
\begin{aligned}
\operatorname*{minimize}_{\theta, u, v} \quad & \sum_{i=1}^{d} u_i \\
\text{subject to} \quad & b - A\theta = v, \\
& -u_i \leq \theta_i \leq u_i, \quad \forall i \in [d], \\
& -n\lambda \leq v_i \leq n\lambda, \quad \forall i \in [d].
\end{aligned}
$$

One can invoke any preferred linear programming solver to solve the above problem. Theoretically, the computational complexity for solving the above linear programming problem with accuracy $\delta$ is $O(d^{\omega+o(1)} \log(d/\delta))$ in expectation for the randomized solver in Cohen et al. (2019) and $O(d^\omega \log^2(d) \log(d/\delta))$ for the deterministic solver in van den Brand (2020), where $\omega$ is the exponent of matrix multiplication complexity with the current bound $\omega \leq 2.371552$ (Williams et al., 2024). Thus, the overall computational complexity of solving the Dantzig selector problem is $O(mnd^2 + d^\omega \log^2(d) \log(d/\delta))$ (for the deterministic solver).

Finally, we remark that in the classical finite-dimensional lasso, elastic net, and Dantzig selector problems, the overall computational complexities are the same as those listed above with $m = 1$. This is because each integral can be replaced with a single scalar multiplication in the pre-processing step.

