# OpenReview forum: "Sparse Contextual CDF Regression"
_TMLR — Accepted by TMLR_

### Review · Reviewer_tKKA · 2024-04-07

**Summary Of Contributions:**

This paper considers the problem of learning a sparse mixture weight of contextual cumulative distribution functions (CDFs). Specifically, data points are generated from a convex combination of $s$ context-dependent CDFs from a set of $d$ known basis functions. This paper adapts three conical regression methods, i.e., LASSO, Elastic Net, and Dantzig Selector, to this new problem and provides the corresponding estimation error upper bounds. It is shown that these $L_1$ based approaches achieve better rates compared to $L_2$ regularized ridge regression. Moreover, a matching information-theoretic lower bound is provided, establishing the minimax optimality of the lasso and Dantzig selector estimators.

**Audience:**

Yes

**Claims And Evidence:**

Yes

**Requested Changes:**

- I noticed that random design results are only provided for LASSO. Could the authors generalize such an analysis to Elastic Net and Dantzig Selector? Or at least describe the challenges of such an extension. I believe such a discussion could be insightful for the readers.

- I felt quite confused reading the first two pages of the paper, but the considered problem became much more apparent after going through the math model on page 3. I think the basic math model of the problem can be moved to section I to improve readability, as the formulation is quite neat.

**Strengths And Weaknesses:**

Strengths:
I enjoyed reading this paper, as all the theoretical results and proofs are presented clearly. Furthermore, the sketch of the proof provided under each theorem is quite helpful in understanding the key technical steps. The setting considered in the paper is general enough for the audience of TMLR.



Weaknesses:

Lack of experiment justification:
Only synthetic results for LASSO under fixed design are provided. Can the authors enlarge the numerical section by including results for LASSO for random design settings and fixed design settings for either Elastic Net or Dantzig Selector?

---

> ### Author Response · Authors · 2024-05-09
>
> Thank you for your interest in reviewing our paper.
>
> **Experiments:** We have added experimental results for the elastic net estimator in the fixed design setting (refer to the updated Figure 1), and experimental results for the ridge, lasso, and elastic net estimators in the random design setting (refer to the new Figure 2). We have updated our GitHub repository with the new code, results, and visualizations for these simulations. We omit simulations for the Dantzig selector due to the computational complexity of solving the constrained optimization problem in Equations 3-4.
>
> **Random design results:** We also have established and added new theoretical results for the elastic net and Dantzig selector estimators in the random design setting (refer to the new Theorems 4 and 6 and the updated Appendices C and D).
>
> **Problem model:** Finally, we have reorganized the paper to present the formal model as part of the introduction (refer to Subsection 1.2), before the related work section.

---

### Review · Reviewer_eGni · 2024-04-10

**Summary Of Contributions:**

The paper studies sparse CDF regression. For Lasso and Dantzig selector formulation, the paper obtains $\sqrt{s \log d/n}$  rate, and for elastic net estimator, the paper obtains $(s^2 \log d / n)^{1/4}$.

**Audience:**

Yes

**Claims And Evidence:**

Yes

**Requested Changes:**

In the abstract, it might be better to clarify the following points:

-	What does s-context dependent CDF mean?

-	When we have $1/\sqrt{n}$ estimators, why should we care $n^{-1/4}$ estimator?

**Strengths And Weaknesses:**

**Strength**

The problem setting sounds interesting, and potentially generalizable to other settings.

**Weakness**

The main weakness of this work is the novelty. The algorithm and analysis seem to solve more-or-less standard sparse regression problems with RIP style conditions. I wonder why the paper had to be specifically about “CDF” regression. Compared to a more well-studied setting of sparse linear regression, can the authors elaborate what is the technical novelty in this work to solve the “CDF” regression? (also, what is the main technical challenge that was not in sparse linear regression?)

---

> ### Author Response · Authors · 2024-05-09
> **Response to eGni (part 1/2)**
>
> Thank you for your interest in reviewing our paper.
>
> **Motivation for CDFs:** We chose to focus specifically on CDF regression in this paper due to the ubiquitous presence of CDFs in statistical applications, the generality of CDFs in representing distributions (as opposed to PDFs or quantiles, which require assumptions for well-definedness), and the well-established utility of CDF estimation in machine learning problems such as multi-armed bandits [1], risk assessment [2], and Markov decision processes [3]. Furthermore, due to the simplicity and foundational nature of CDFs as a mathematical building block, we expect that our work will be of future interest to many hitherto unknown applications in finance and machine learning. We remark that prior accepted work in TMLR has studied the contextual CDF regression problem in the dense setting [4], but our present work on sparse CDF regression allows for "feature selection" which means it is applicable in broader settings as it removes the requirement for prior knowledge of a suitable CDF basis. (These points are discussed in Sections 1 and 2 of our paper.)
>
> **Novelty:** In our paper, we analyze sparse regression and feature selection methods in the functional setting where both the features and responses are CDFs, as opposed to the classical finite-dimensional setting. This change in perspective requires some technical novelty. For example, the definition of the Gramian matrix $U_n$ in terms of an integral over the support of the underlying random variable of the CDFs must necessarily differ from finite-dimensional problems. Likewise, the definitions of the restricted eigenvalue, restricted isometry, and restricted orthogonality properties also commensurately change in this functional setting, and we analytically derive a high-probability bound on the inner products between the sampling errors and basis functions (Lemma 1) using intrinsic properties of CDFs without introducing discretization error in our theoretical analysis. Moreover, we note that one limitation of known estimation error upper bounds for classical finite-dimensional sparse regression (see, e.g., Equation (7.26) in [5]) is that they depend on the magnitudes of the features and the noise variance. In contrast, for our proposed sparse CDF regression model and estimator, the estimation error upper bound is independent of response and noise scales. This technical difference in the bounds stems from utilizing the CDF assumption in our arguments.
>
> Notwithstanding these points, we highlight that TMLR guidelines explicitly mention that interest, as opposed to novelty, should form the basis for evaluation, as stated in https://jmlr.csail.mit.edu/tmlr/reviewer-guide.html:
>
> > Crucially, it should not be used as a reason to reject work that isn't considered "significant" or "impactful" because it isn't achieving a new state-of-the-art on some benchmark. Nor should it form the basis for rejecting work on a method considered not "novel enough", as novelty of the studied method is not a necessary criteria for acceptance. We explicitly avoid these terms ("significant", "impactful", "novel"), and focus instead on the notion of "interest". If the authors make it clear that there is something to be learned by some researchers in their area from their work, then the criteria of interest is considered satisfied.
>
> Due to the wide-ranging applicability of CDFs and the technical novelty in our derivations discussed above, we believe that our work is interesting to the machine learning community and may be helpful in deriving results in downstream applications.
>
> **Context-dependent CDF:** For clarity, we have changed "context dependent" to "context-dependent" throughout the paper (including the abstract). The terminology "contexts" is often used in the bandits and reinforcement learning community. A context-dependent CDF is a bivariate function $F(x, t)$ such that for any value of the *context variable* $x$, the mapping $t \mapsto F(x, t)$ is a CDF, as explained in Section 1.2. In the abstract, we meant that we consider $s$ different context-dependent CDFs; this should be clearer now with the hyphenation.

---

> ### Author Response · Authors · 2024-05-09
> **Response to eGni (part 2/2)**
>
> **$\tilde{O}(n^{-1/4})$ estimator:** The elastic net estimator can achieve either a $\tilde{O}(n^{-1/2})$ or a $\tilde{O}(n^{-1/4})$ error bound depending on the assumptions imposed. We mention that $\tilde{O}(n^{-1/4})$ error bound because it may be utilized without any assumptions on the Gramian matrix $U_n$, whereas the lasso estimator requires a non-trivial restricted eigenvalue condition ($\kappa > 0$) on $U_n$. Imposing a $\kappa > 0$ condition recovers an $\tilde{O}(n^{-1/2})$ scaling for elastic net, similar to lasso. To emphasize this, we have modified Theorem 3 to present the optimal value of $\lambda_2$ in terms of both $\kappa$ and $n$, and we have added the following discussion below the statement of Theorem 3:
>
> > We emphasize that Theorem 3 produces non-trivial estimation bounds even when no assumptions are placed on $U_n$ (i.e., $\kappa = 0$), in which case the bound in (5) has $\tilde{O}(\sqrt{s} / \sqrt[4]{n})$ complexity. However, when $\kappa > 0$ and the $\ell^2$-regularization hyperparameter is sufficiently small with $\lambda_2 = O(1 / \sqrt{n})$, the elastic net estimation bound exhibits $\tilde{O}(\sqrt{s/n})$ scaling, comparable with the result for the lasso estimator in Theorem 1.
>
> [1]: Asaf Cassel, Shie Mannor, and Assaf Zeevi. A general framework for bandit problems beyond cumulative objectives. *Mathematics of Operations Research*, 2023.
>
> [2]: Audrey Huang, Leqi Liu, Zachary Lipton, and Kamyar Azizzadenesheli. Off-policy risk assessment in contextual bandits. *Advances in Neural Information Processing Systems*, 34, 2021.
>
> [3]: Audrey Huang, Leqi Liu, Zachary C Lipton, and Kamyar Azizzadenesheli. Off-policy risk assessment for Markov decision processes. In *Artificial Intelligence and Statistics*, 2022.
>
> [4]: Qian Zhang, Anuran Makur, and Kamyar Azizzadenesheli. Functional linear regression of cumulative distribution functions. *Transactions on Machine Learning Research*, 2024. URL https://openreview.net/forum?id=ZOqJCP4eMk.
>
> [5]: Martin J Wainwright. *High-dimensional statistics: A non-asymptotic viewpoint*, volume 48. Cambridge University Press, 2019.

---

### Review · Reviewer_G4mk · 2024-04-29

**Summary Of Contributions:**

This paper investigates sparse contextual CDF regression using several newly proposed estimators. A contextual CDF is a parameterized cumulative distribution function (CDF) where an additional input, known as the "context", can vary the form of the CDF. To manage the complexity, the authors limit the number of potential contextual CDFs by employing a fixed finite basis, making the problem theoretically approachable.

In this scenario, the authors posit an unknown true contextual CDF that is a (sparse) convex combination of $d$ known base CDFs. For each observation $j \in [n]$, a context $x^{(j)} \in \mathcal{X}$ is chosen either through fixed design or randomly sampled from an unknown distribution (in random design). Conditioned on the context, a sample is obtained from the convex combination. The objective of the paper is to recover the coefficients of this convex combination based on the $n$ observed samples, with the coefficients assumed to be $s$-sparse.

The authors introduce three estimators for this purpose: i) the Lasso estimator, ii) Elastic Net estimator, and iii) the Dantzig selector. Assuming a geometric regularity condition on the basis CDFs (termed as the Restricted Eigenvalue condition in Definition 1, similar to RIP), the authors provide upper-bound estimates for the error across all scenarios, complemented by some lower-bound analyses. The upper-bound results, akin to conventional sparse regression, indicate an error reduction rate of $(s/n)^{1/2}$ rather than $(d/n)^{1/2}$, demonstrating how leveraging sparsity as a structural property can diminish the required sample size, as anticipated.

The paper includes various numerical simulations towards the end, although I have not validated these results myself.

**Audience:**

Yes

**Broader Impact Concerns:**

none.

**Claims And Evidence:**

Yes

**Requested Changes:**

Please see the Weaknesses section.

**Strengths And Weaknesses:**

**Strengths**:

- The paper is exceptionally well-written and easy to follow.

- The problem addressed in this work is relevant and appealing to a wide audience within the machine learning community.

- The mathematical derivations are precise and rigorous, based on my review. I did not identify any significant mathematical errors.

- The paper proposes and analyzes several estimators, providing a comprehensive investigation into the topic.

-------------------------

**Weaknesses**:

- The primary weakness is that the contribution of this work appears somewhat incremental. While the combination of contextual CDF estimation and sparse regression is novel, the individual ideas are not surprising or groundbreaking. The results align with expectations.

- The proofs are relatively straightforward and follow standard procedures from (for example) Wainwright's works combined with prior recent works on CDF regression, which is understandable given the similarity between sparse regression and sparse contextual CDF regression. The same simplicity extends to the lower-bound proof, which basically is a slight varaition of the bound given by Zhang et al. (also outlined by the authors).

- Theorem 2, which presents the error upper-bound for random design, has a high sample complexity. Although the error decreases as $\mathcal{O}\left((s/n)^{1/2}\right)$, the requirement that $n$ be greater than $\mathcal{O}\left(d^2\right)$ contradicts the claimed sample complexity improvement at the paper's outset. This issue primarily affects the random design and not fixed design scenarios.

- Proposing the Restricted Eigenvalue Condition as an analogue of RIP for sparse contextual CDF regression is a natural and promising approach. However, it would enhance the paper if the authors could provide converse results to demonstrate that this condition is not only sufficient but also necessary for the problem at hand.

- The paper lacks a detailed discussion on the computational complexity of the proposed estimators. It would be beneficial for the authors to address this aspect, especially in comparing the computational demands of CDF regression versus ordinary sparse regression as found in existing literature.

---

> ### Author Response · Authors · 2024-05-09
> **Response to G4mk (part 1/2)**
>
> Thank you for your interest in reviewing our paper.
>
> **Weaknesses 1 and 2:** As we mention in the response to reviewer eGni, our paper analyzes sparse regression and feature selection methods in the functional setting where both the features and responses are CDFs, as opposed to the classical finite-dimensional setting. This change in perspective requires some technical novelty. For example, the definition of the Gramian matrix $U_n$ in terms of an integral over the support of the underlying random variable of the CDFs must necessarily differ from finite-dimensional problems. Likewise, the definitions of the restricted eigenvalue, restricted isometry, and restricted orthogonality properties also commensurately change in this functional setting, and we analytically derive a high-probability bound on the inner products between the sampling errors and basis functions (Lemma 1) using intrinsic properties of CDFs without introducing discretization error in our theoretical analysis. Moreover, we note that one limitation of known estimation error upper bounds for classical finite-dimensional sparse regression (see, e.g., Equation (7.26) in [3]) is that they depend on the magnitudes of the features and the noise variance. In contrast, for our proposed sparse CDF regression model and estimator, the estimation error upper bound is independent of response and noise scales. This technical difference in the bounds stems from utilizing the CDF assumption in our arguments.
>
> Moreover, as you noted, the specific setting of sparse contextual CDF regression under investigation is novel and broadly applicable to problems of interest in machine learning. We highlight that the TMLR guidelines for reviewers explicitly mention that interest, as opposed to novelty, should form the basis for a reviewer decision. Viewed through this lens, we regard results aligning with a priori expectations as an appealing quality of our work, showing that principled approaches to solving finite-dimensional sparse regression problems remain applicable in the context of CDFs. As stated in https://jmlr.csail.mit.edu/tmlr/reviewer-guide.html,
>
> > Crucially, it should not be used as a reason to reject work that isn't considered "significant" or "impactful" because it isn't achieving a new state-of-the-art on some benchmark. Nor should it form the basis for rejecting work on a method considered not "novel enough", as novelty of the studied method is not a necessary criteria for acceptance. We explicitly avoid these terms ("significant", "impactful", "novel"), and focus instead on the notion of "interest". If the authors make it clear that there is something to be learned by some researchers in their area from their work, then the criteria of interest is considered satisfied.
>
> Due to the wide-ranging applicability of CDFs and the technical novelty in our derivations discussed above, we believe that our work is interesting to the machine learning community and may be helpful in deriving results in downstream applications.
>
> **Weakness 3 (sample complexity):** We have changed the phrase "sample complexity" to "estimation error" in the abstract to better characterize our results; the scalings presented at the outset were not sample complexities, but estimation error bounds. We also clarify that the statement of Theorem 2 is non-vacuous and non-trivial despite requiring $n = \Omega(d^2)$ samples, because the actual magnitude of the estimation error bound $\tilde{O}(\sqrt{s/n})$ for lasso regression is an improvement over the $\tilde{O}(\sqrt{d/n})$ bound for ridge regression. This improvement remains valid even when $n$ is larger than $d^2$.

---

> ### Author Response · Authors · 2024-05-09
> **Response to G4mk (part 2/2)**
>
> **Weakness 4 (necessity of RIP):** To our knowledge, even in the literature on classical finite-dimensional sparse regression, there are no unified RIP-like necessary conditions for the lasso, elastic net, and Dantzig selector estimators. Furthermore, in the special case of finite-dimensional lasso, prior works [1] and [2] have developed morally RPI-like conditions, e.g., "irrepresentability", that are *almost* necessary and sufficient for the lasso estimator. However, such conditions are not known to be entirely necessary. In a different vein, it is worth mentioning that while restricted eigenvalue or RIP-like conditions yield $\tilde{O}(\sqrt{s/n})$ estimation error bounds for finite-dimensional lasso, weaker bounds on in-sample risk can be obtained under almost no assumptions on feature matrices (this complementary direction is not the focus of our work). We acknowledge the importance of investigating necessary conditions, but we feel this is a separate open problem for all three estimators we propose. At present, we do not have any concrete approach to develop unified necessary conditions for our three estimators (or even the finite-dimensional versions of these estimators). So, we have added to the Conclusion section of our manuscript that finding necessary conditions for our proposed estimators is an important direction for future work.
>
> **Weakness 5 (computational complexity):** The estimators we propose are objective formulations, not specific algorithms for optimizing said formulations, so the notion of computational complexity is not immediately applicable to this context. Nonetheless, we have added a discussion on the computational complexity of typical numerical implementations of our estimators and classical finite-dimensional estimators using specific algorithms in Appendix G. There, we discuss both the cost of approximating the integrals in our formal model and the complexity of running existing iterative solvers for the optimization problems. We refer the reviewer to Appendix G for further details.
>
> [1]: Peng Zhao and Bin Yu. On model selection consistency of lasso. *Journal of Machine Learning Research*, 7: 2541–2563, 2006.
>
> [2]: Nicolai Meinshausen and Bin Yu. Lasso-type recovery of sparse representations for high-dimensional data. *The annals of statistics*, 37(1), 2009.
>
> [3]: Martin J Wainwright. *High-dimensional statistics: A non-asymptotic viewpoint*, volume 48. Cambridge University Press, 2019.

---

### Decision · Action_Editor_yQ4W · 2024-06-22

**Recommendation:** Accept as is

**Comment:**

This paper studies sparse contextual CDF regression, and provides convergence rate analysis for Lasso, elastic net, and Dantzig selector estimators. The results match the corresponding error bounds for finite-dimensional regression. Furthermore, the Lasso and Dantzig selector estimators are shown to be minimax optimal (up to logarithmic factors). The revised version clarifies a few technical questions raised in the reviews, and also elaborates technical differences from the finite-dimensional case. While there are concerns over significance, the paper contains useful contributions, is well-written, and should be accepted to TMLR.

**Audience:**

The paper is interesting to communities in statistics and learning theory.

**Claims And Evidence:**

This paper studies sparse contextual CDF regression, and provides convergence rate analysis for three estimators. The main contributions are theoretical, and are supported by full proofs. The paper also evaluates their empirical performance through numerical simulations.